



# Long-term observational constraints of organic aerosol dependence on inorganic species in the southeast US

Yiqi Zheng[1,2,*], Joel A. Thornton[3], Nga Lee Ng[4,5,6], Hansen Cao[7], Daven K. Henze[7], Erin E. McDuffie[8,9], Weiwei Hu[10,11], Jose L. Jimenez[11], Eloise A. Marais[12], Eric Edgerton[13], Jingqiu Mao[1,2,*]

[1]Geophysical Institute, University of Alaska Fairbanks, Fairbanks, AK, USA
[2]Department of Chemistry and Biochemistry, University of Alaska Fairbanks, Fairbanks, AK, USA
[3]Department of Atmospheric Sciences, University of Washington, Seattle, WA, USA
[4]School of Chemical and Biomolecular Engineering, Georgia Institute of Technology, Atlanta, GA, USA
[5]School of Earth and Atmospheric Sciences, Georgia Institute of Technology, Atlanta, GA, USA
[6]School of Civil and Environmental Engineering, Georgia Institute of Technology, Atlanta, GA, USA
[7]Department of Mechanical Engineering, University of Colorado Boulder, Boulder, CO, USA
[8]Department of Physics and Atmospheric Science, Dalhousie University, Halifax, Nova Scotia, Canada
[9]Department of Energy, Environmental & Chemical Engineering, Washington University in St. Louis, St Louis, MO, USA
[10]State Key Laboratory of Organic Geochemistry, Guangzhou Institute of Geochemistry, Chinese Academy of Science (CAS), Guangzhou, China
[11]Department of Chemistry and CIRES, University of Colorado Boulder, Boulder, CO, USA
[12]School of Physics and Astronomy, University of Leicester, Leicester, LE1 7RH, UK
[13]Atmospheric Research & Analysis, Inc., Cary, NC, USA

[*]Correspondence to: Yiqi Zheng (yzheng4@alaska.edu) and Jingqiu Mao (jmao2@alaska.edu)

**Keywords:** Secondary organic aerosol; IEPOX; isoprene; coating; aerosol acidity.



# 1 **Abstract**

Organic aerosol (OA), with a large biogenic fraction in summertime southeast US, adversely
impacts on air quality and human health. Stringent air quality controls have recently reduced
anthropogenic pollutants including sulfate, whose impact on OA remains unclear. Three filter
measurement networks provide long-term constraints on the sensitivity of OA to changes in
inorganic species, including sulfate and ammonia. The 2000-2013 summertime OA decreases by
1.7~1.9%/year with little month-to-month variability, while sulfate declines rapidly with
significant monthly difference in early 2000s. In contrast, modeled OA from a chemical-
transport model (GEOS-Chem) decreases by 4.9%/year with much larger monthly variability,
largely due to the predominant role of acid-catalyzed reactive uptake of epoxydiols (IEPOX)
onto sulfate. The overestimated modeled OA dependence on sulfate can be improved by
implementing a coating effect and assuming constant aerosol acidity, suggesting the needs to
revisit IEPOX reactive uptake in current models. Our work highlights the importance of
secondary OA formation pathways that are weakly dependent on inorganic aerosol in a region
that is heavily influenced by both biogenic and anthropogenic emissions.



## 1. Introduction

Organic aerosol (OA) accounts for a large fraction of ambient fine particulate matter mass,
which strongly affects air quality, regional climate, and human welfare (Jimenez et al., 2009).
Since the implementation of the Clean Air Act Amendments of 1990, there has been a significant
decline in ambient aerosol in the United States, mostly due to reductions in inorganic aerosol
mass following changes in emissions of sulfur dioxide ($SO_2$), nitrogen oxides ($NO_x = NO + NO_2$),
as well as reductions in black carbon (EPA, 2011), leaving OA as the major component of fine
particulate matter (50~70%) over the southeast US, especially in summer (Attwood et al., 2014;
Kim et al., 2015). OA can be directly emitted by combustion processes (primary organic aerosol,
POA) or secondarily formed (SOA) from the atmospheric oxidation of biogenic volatile organic
compounds (BVOCs), mainly isoprene and monoterpenes, and also precursors from
anthropogenic sources and biomass burning (Hayes et al., 2015; Hodshire et al., 2019). OA has
also been declining across much of the US over the past few decades, primarily due to decreased
anthropogenic emissions from vehicle and residential fuel-burning, except for the southeast US
(Ridley et al., 2018). The southeast US is one of the largest BVOC emission hotspots in the
world (Guenther et al., 2006), and at the same time is heavily populated with large anthropogenic
emissions of pollutants. Biogenic SOA may account for 60-100% of OA in summertime
southeast US (Kim et al., 2015; Xu et al., 2015b). To what extent biogenic SOA could be
mediated through emission control strategies remains an open question (Carlton et al., 2010;
Mao et al., 2018).
The oxidation of BVOCs produces hundreds of intermediate products. Some products have low
volatility that can partition onto the condensed phase, while some gas-phase products can react in





the aqueous phase to form SOA. SOA formed from uptake of isoprene epoxydiols (IEPOX-
SOA) (Paulot et al., 2009) appears to be the major confirmed aqueous SOA product globally,
being important in all high isoprene and lower NO regions (Hu et al., 2015), along with glyoxal
formed from isoprene and aromatics (Fu et al., 2008). Formation of SOA in clouds was
investigated in the southeast US and found to be not statistically significant (Wagner et al.,
2015). These pathways have been implemented into 3-dimensional global atmospheric chemistry
and climate models using two different approaches. First, to simulate the partitioning of organic
vapors, the BVOC oxidation products can be grouped based on their volatility (Volatility Basis
Set, VBS), and the product yields and vapor pressures are parameterized for each surrogate
precursor (Donahue et al., 2006; Pankow, 1994). Such empirical VBS schemes are usually
derived using dry laboratory chamber experiments (with relative humidity RH<10%) and do not
explicitly depend on aerosol water, RH, or inorganic aerosol mass or composition. Therefore,
here we refer to the SOA formed through partitioning calculated by VBS as dry SOA. Second, a
more explicit representation of aqueous SOA formation from isoprene products has been used
recently, which incorporates dependence on inorganic aerosol volume and aerosol acidity
(Budisulistiorini et al., 2017; Ervens et al., 2011; Fu et al., 2008; Marais et al., 2016; Pye et al.,
2013). The relative contribution of dry versus aqueous SOA to total OA mass in the atmosphere
is uncertain and has limited observational constraints.

Long-term field measurements show a decreasing trend of OA in the southeast US (Attwood et
al., 2014; Hidy et al., 2014; Kim et al., 2015), which is likely linked to reductions in
anthropogenic POA and SOA (Blanchard et al., 2016; Ridley et al., 2018), sulfate (Blanchard et
al., 2016; Malm et al., 2017; Marais et al., 2017; Xu et al., 2015b, 2016) and $NO_x$ (Carlton et al.,




2010; Pye et al., 2010, 2019a; Xu et al., 2015b). The influence of sulfate on OA is thought to be
mainly due to its influence on the uptake of isoprene gas-phase oxidation products, which are
often small molecules that cannot directly condense due to high vapor pressure, but may undergo
aqueous-phase reactive uptake onto wet sulfate particles to form aqueous SOA, as suggested by
extensive laboratory and field studies (Budisulistiorini et al., 2015; Hu et al., 2015; Li et al.,
2016; Liggio et al., 2005; McNeill et al., 2012; Riedel et al., 2016; Shrivastava et al., 2017;
Surratt et al., 2010; Tan et al., 2012; Xu et al., 2016, 2015b). $NO_x$ plays a complex role in
regulating oxidation capacity and different oxidation pathways (Kiendler-Scharr et al., 2016;
Kroll et al., 2005, 2006; Li et al., 2018; Ng et al., 2017; Presto et al., 2005; Shrivastava et al.,
2019; Zheng et al., 2015; Ziemann and Atkinson, 2012). Prior 3-D modeling studies with
different SOA mechanisms provide different explanations for the long-term OA trend observed
in the southeast US. For example, the dry SOA calculated by VBS framework with $NO_x$-
dependent yields implies a small decrease in OA following the reductions of $NO_x$ (Pye et al.,
2013; Zheng et al., 2015), but has little dependence on changes in inorganic aerosol mass such as
sulfate. On the other hand, models using aqueous SOA formation from isoprene attributed the
decreasing OA from 1991 to 2013 to reductions in sulfate (Marais et al., 2017) but showed
greater interannual variability than was observed. The driving mechanism for the OA trend in the
southeast US remains to be elucidated.

Here we use observations from three surface filter-based networks (IMPROVE, SEARCH,
CSN), combined with a 3-dimensional chemical transport model GEOS-Chem v12.1.0, to
examine the long-term trend and more importantly, the month-to-month variability of OA in the





southeast US during 2000-2013. The results provide new observational constraints on the drivers
of OA variability and the SOA formation mechanisms in the southeast US.

## 2. Methods

**2.1 Observational datasets.**
We use surface filter-based measurement of fine particulate matter mass and composition
(including organic carbon, OC) in 2000-2013 from three networks: the Interagency Monitoring
of Protected Visual Environments (IMPROVE) (Solomon et al., 2014), the SouthEastern Aerosol
Research and Characterization (SEARCH) (Edgerton et al., 2005), and the Environmental
Protection Agency's $PM_{2.5}$ National Chemical Speciation Network (CSN) (Solomon et al.,
2014). We select 21 IMPROVE sites, 3 SEARCH rural sites and 36 CSN sites within the
southeast US region [29°~37°N, 74°~96°W] (Figure S1). The SEARCH sites are organized in
rural/urban pairs (Edgerton et al., 2005) and only the data from the rural sites are used here to
represent background conditions. IMPROVE sites are mostly rural (Solomon et al., 2014). The
OC measurement in the CSN network in 2004-2009 gradually shifted to a different protocol and
analytical technique than the early 2000s, which led to the discontinuity in long-term trend
(Figure S2), therefore we only use CSN data to examine the monthly variability of OA, and
focus on IMPROVE and SEARCH for all analysis. The 3-day OC measurement from IMPROVE
and daily OC from SEARCH and CSN are averaged to monthly values. A factor of 2.1 is used to
convert measured organic carbon (OC) to organic aerosol mass, as suggested by the southeast
US field measurements (Pye et al., 2017; Schroder et al., 2018).



We use OA measurements by Aerosol Mass Spectrometer (AMS) from the Southern Oxidant
and Aerosol Studies campaign (SOAS) at the Centerville, AL Site in 06/01/2013-07/15/2013
(SOAS2013). The OA measurements and derived IEPOX-SOA factor calculated by Positive
Matrix Factorization (PMF) analysis (Hu et al., 2015; Xu et al., 2015a, 2018) are from two
independent groups: one group from Georgia Institute of Technology led by Prof. Nag Lee Ng,
the other from University of Colorado Boulder led by Prof. Jose L. Jimenez, denoted as Obs_GT
and Obs_CU, respectively.

**2.2 Modeling framework**
**2.2.1 GEOS-Chem**
In this study we use the 3-dimensional global chemical transport model GEOS-Chem version
12.1.1 (DOI: 10.5281/zenodo.2249246, https://github.com/geoschem/) with detailed $O_3$-$NO_x$-
$HO_x$-CO-VOC-aerosol tropospheric chemistry (Bey et al., 2001; Mao et al., 2013). Isoprene
chemistry is described in (Fisher et al., 2016; Travis et al., 2016). GEOS-Chem is driven by
offline meteorology 1999-2013 from the NASA Modern-Era Retrospective analysis for Research
and Applications, version 2 (MERRA-2 https://gmao.gsfc.nasa.gov/reanalysis/MERRA-2/). The
global anthropogenic emissions are from the Community Emissions Data System (CEDS)
inventory, with the US region replaced by the EPA's National Emission Inventory for 2011
(NEI11v1). The monthly mean anthropogenic emissions of CO, $SO_2$, $NO_x$, $NH_3$, VOCs, OC and
black carbon are mapped over 0.1°×0.1° and scaled to the year 2011 by the ratio of national
annual totals from 2000 to 2013 (Travis et al., 2016). Biomass burning emissions are from
Global Fire Emissions Database version 4 (GFED4) (Randerson et al., 2015). Biogenic
emissions of isoprene and terpenes are online calculated by the Model of Emissions of Gases and



Aerosols from Nature (MEGAN2.1) (Guenther et al., 2012) that is also driven by MERRA-2
meteorology.

For organic aerosol, we employ the complexSOA scheme for SOA modeling for all simulations
in this study (Marais et al., 2016; Pye et al., 2010). POA are regarded as nonvolatile. This SOA
modeling includes a 4-product Volatility-Basis-Set (VBS) for SOA formation from reversible
condensation of oxidation products of biogenic terpenes (including monoterpenes and
sesquiterpenes), and anthropogenic VOCs, referred to as terpene-SOA and anthropogenic SOA,
respectively. The SOA calculated through VBS parameterization is fitted based on dry chamber
(RH<10%) results independent of inorganic aerosol, aerosol water and RH (Pye et al., 2010).
The complexSOA scheme also includes aqueous SOA formed from reactive uptake of isoprene
oxidation products, including IEPOX, glyoxal, $C_4$ epoxides, methylglyoxal, non-IEPOX product
of the ISOPOOH oxidation and hydroxynitrates from $NO_3$-initiated oxidation (Marais et al.,
2016). The sulfate-nitrate-ammonium aerosol thermodynamics including aerosol acidity is
computed with the ISORROPIA II thermodynamic model (Fountoukis and Nenes, 2007; Pye et
al., 2009; Song et al., 2018).

We run the GEOS-Chem model at 4°×5° latitude by longitude continuously from 10/01/1999 to
12/31/2013. For each year, the restart file at 05/01 from the continuous 4°×5° simulation has
been regridded to 2°×2.5° and is used to initiate 2°×2.5° simulations from 05/01 to 08/31 each
year. The 2°×2.5° simulations are adequate when modeling continental boundary layer chemistry
(Yu et al., 2016). The May results are discarded as spin-up and the results of June, July and
August are used for analysis. We do four sets of 2°×2.5° simulations: Default (using default





complexSOA scheme); CT (with coating effect for IEPOX reactive uptake); CT_newNH$_3$ (with
coating effect and US NH$_3$ emissions replaced by satellite-derived NH$_3$ inventory); CT_H01
(with coating effect and fixing aerosol a$_{H+}$ at 0.1 mol/L when calculating IEPOX reactive
uptake).

**2.2.2 Coating**
The default IEPOX-SOA mechanism in GEOS-Chem uses aerosol-phase reaction rates from
laboratory chamber studies with pure acidic inorganic particles (Gaston et al., 2014; Riedel et al.,
2015), and a representative effective Henry's law constant obtained by matching the model to the
observations at SOAS2013 campaign, to estimate the reactive uptake coefficient $\gamma_{IEPOX}$. In the
default scheme, $\gamma_{IEPOX}$ is calculated as follows:
$$\frac{1}{\gamma_{IEPOX}} = \frac{R_p \omega}{4 D_g} + \frac{1}{\alpha} + \frac{1}{\Gamma_{aq}}$$

$$\Gamma_{aq} = \frac{4 V R T H_{aq} k_{aq}}{S_a \omega}$$

Where $R_p$ is the particle radius of the inorganic sulfate-nitrate-ammonium particle (cm), $\omega$ is the
mean molecular speed (cm/s), $D_g$ is the gas-phase diffusion coefficient (0.1 cm$^2$/s), $\alpha$ is the mass
accommodation coefficient ($\alpha$=0.1), $S_a$ is the total (wet) particle surface area (cm$^2$/cm$^3$), $V$ is the
total (wet) particle volume (cm$^3$/cm$^3$), $R$ is the ideal gas constant (L atm/mol/K), $T$ is temperature
(K), $H_{aq}$ is the Henry's law coefficient (1.7×10$^7$ M/atm), and $k_{aq}$ is the first-order reaction rate
constant (s$^{-1}$):
$$k_{aq} = k_{H^+}[H^+] + k_{nuc}[nuc]a_{H^+} + k_{ga}[ga]$$

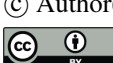



where $k_{H^+}$ (=0.036 M$^{-1}$s$^{-1}$), $k_{nuc}$ (=2×10$^{-4}$ M$^{-1}$s$^{-1}$) and $k_{ga}$ (=7.3×10$^{-4}$ M$^{-1}$s$^{-1}$) are the reaction
rates due to acid-catalyzed ring-opening, presence of nucleophiles (including nitrate and sulfate)
and presence of bisulfate acids, respectively (Gaston et al., 2014; Marais et al., 2016).

We implement a linear coating effect for the IEPOX-SOA formation. The coating effect is fitted
using laboratory-derived values of $\gamma_{IEPOX}$ on particles containing both ammonium bisulfate and
ethylene glycol under RH=50% conditions (Gaston et al., 2014). In the coating scheme, $\gamma'_{IEPOX}$
is calculated as above with $R_p$, $V$ and $S_a$ updated considering OA coated outside the inorganic
core. Then, the fitted function is applied to modify $\gamma'_{IEPOX}$:
$$\gamma_{IEPOX\_modified} = \gamma'_{IEPOX} \times (1 - 1.3 \times \chi_{org})$$
where $\chi_{org}$ is the mass fraction of OA in the mixed particle including both the inorganic aerosol
and OA. When $\chi_{org}$>0.7, the IEPOX uptake will be terminated, i.e. $\gamma_{IEPOX\_modified} = 0$. We
assume all OA is coated outside the inorganic aerosol core when calculating the IEPOX reactive
uptake. The increased particle radius $R_p$ and surface area $S_a$ of the mixed particle will partially
offset the impact of reduced reaction probability $\gamma_{IEPOX\_modified}$, consistent with another study
(Jo et al., 2019).

**2.2.3 Satellite-derived NH₃ emissions**
We use the Cross-track Infrared Sounder (CrIS) satellite-derived NH$_3$ emissions in a sensitivity
test in this study. The top-down monthly NH$_3$ emissions over the contiguous US at 0.25° ×
0.3125° latitude by longitude are derived from CrIS v1.5 measurements of NH$_3$ profiles
(Shephard and Cady-Pereira, 2015) for the year 2014 through a 4D-Var approach using GEOS-
Chem and its adjoint model (Henze et al., 2007). The CrIS-derived emissions are then regridded



to 0.1°×0.1° to replace the default NEI11 emissions for the year 2011 and applied the same
annual scaling factors for 2000-2013. There is no significant trend from 2000 to 2013 (Figure
S3), consistent with other studies suggesting nearly constant $NH_3$ emissions from 2001 to 2014
(Butler et al., 2016). The CrIS-derived emissions used the HTAPv2 emissions inventory as the
prior emissions, which is based on the 2008 NEI emissions over the US (Janssens-Maenhout et
al., 2015). The top-down annual mean emissions are ~52% higher than the prior emissions, likely
because the prior emissions underestimate agricultural emissions, in particular springtime
fertilizer and livestock sources over the Central US. Meanwhile, some smaller values then the
prior emissions were found in the Central Valley, southern Minnesota, northern Iowa and
southeast North Carolina during warm months. Using the top-down emissions in GEOS-Chem
increases the correlation coefficient (*r*) between modeled monthly mean $NH_3$ and surface
observations from 0.74 to 0.93 and reduces the normalized mean bias of domain-averaged annual
mean simulated $NH_3$ by a factor of 1.9. The seasonal cycle of simulated wet $NH_4^+$ is also
improved (*r* increased from 0.70 to 0.86), but the normalized mean bias of domain-averaged
annual simulated wet $NH_4^+$ increases from 0.34 to 0.96 due to overly strong wet scavenging in
the model. The latter issue was ultimately resolved in Cao et al. (submitted) and the final top-
down emissions reported therein differ from those reported here; nevertheless, the emissions
estimates used here provide a valuable basis for conducting a sensitivity experiment.

**2.3 Multivariate linear regression analysis**
In this study we did a multivariate regression analysis of monthly IEPOX-SOA ($\mu g/m^3$) against
sulfate aerosol ($\mu g/m^3$), aerosol acidity $a_{H+}$ (mol/L) and isoprene emission ($ISOP_{emis}$ mg/m²/hr):
$$IEPOX\text{-}SOA = \beta_1 \times sulfate + \beta_2 \times a_{H^+} + \beta_3 \times ISOP_{emis} + b$$





Mean values have been subtracted from all variables, which are then divided by standard
deviations. $\beta_1$, $\beta_2$ and $\beta_3$ are standardized partial regression coefficients associated with sulfate
aerosol, $a_{H+}$ and isoprene emission, and can be directly compared to evaluate the relative
importance of the three variables. We apply the regression analysis using monthly data within
different time frames (2000-2013, 2000-2004, 2005-2008 and 2009-2013 as in Table S1) to
determine the evolving importance of variables.


**3. Results**
**3.1 Long-term trend and month-to-month variability (MMV) of OA**
In the southeast US, observations from the IMPROVE and SEARCH network both show a
reduction in summertime surface OA concentration from 2000 to 2013 (Figure 1). Observational
results are averaged using 21 IMPROVE sites and 3 SEARCH sites within the southeast US. OA
concentration averaged over June-July-August (JJA) 2000-2013 is 4.2 μg/m³ from the
IMPROVE sites, and 5.7 μg/m³ from SEARCH sites. A similar ~30% summertime low bias on
the IMPROVE sites was documented by Kim et al. (2015) compared to the SEARCH sites,
which is thought to be due to evaporation of OA from the filters after collection, as the
IMPROVE filters stay several days on site after sampling and are shipped without refrigeration,
while the SEARCH filters are analyzed in-situ. Despite different magnitudes, OA from the two
networks demonstrate similar trends and interannual variability. The 2000-2013 trend of JJA OA
mass is -1.7%/year for IMPROVE and -1.9%/year for SEARCH. Compared to the slow decrease
in OA, a faster declining trend is found for sulfate from IMPROVE (-6.9%/year) and SEARCH
(-6.7%/year) for the same period.





Compared to the observations, the default GEOS-Chem model predicts a steeper decreasing
trend of OA mass during 2000-2013 (Figure 1). Modeling results are averaged over the domain
[29°~37°N, 74°~96°W] excluding ocean grid cells (Figure S1). The 2000-2013 JJA-averaged
OA from the default model is 6.7 $\mu g/m^3$, higher than OA from IMPROVE and SEARCH.
Modeled total OA mass decreases at a rate of 4.9%/year, about 1.9 (1.6) times faster than
IMPROVE (SEARCH) OA (student's t-test p<0.001). The strong reduction in total OA mass is
dominated by aqueous SOA, especially through reactive uptake of IEPOX, with no decreasing
trend in other components (Figure 1). The contribution of IEPOX-SOA to total OA mass
decreases from 61% in the early 2000s to 28% in 2013.

A main constraint comes from the MMV of OA in the southeast US. IMPROVE and SEARCH
OA observations show little variability among June, July and August, despite large MMV of
sulfate in early 2000s (Figure 2A). We find similar behavior from another observation network,
CSN. The discontinuity in OA trend in the CSN network is due to different protocols applied
(Figure S2). Within sites using the same protocol, there are no systematic monthly differences,
which agrees with IMPROVE and SEARCH. In contrast, modeled OA displays large MMV
between June, July and August from 2000 to 2008, where OA in July and August is 1~3 times of
June values (Figure 2A). Such large MMV is dominated by aqueous SOA, especially from the
reactive uptake of IEPOX. Prior to 2008, the simulated IEPOX-SOA alone can be up to a factor
2 higher than the observed total OA (Figure 2). The other components including POA and dry
SOA (including terpene-SOA and Anthropogenic SOA) formed through partitioning together
have low concentrations and small MMV. The default model well captures the variability of



observed sulfate (Figure 2A), with an average of 3.8 μg/m$^3$ and a trend of -6.9%/year, as
compared to -6.9%/year (average concentration 4.2 μg/m$^3$) from IMPROVE and -6.7%/year
(average concentration 4.3 μg/m$^3$) from SEARCH.

The large MMV in the model suggests a much stronger modeled OA dependence on sulfate than
observations. In 2000-2004, changes in modeled sulfate from June to July and/or August
correspond to large MMV of modeled OA mass. In contrast, little MMV is found in observed
OA mass during the same months despite large MMV in observed sulfate (Figure 2A). From a
linear regression analysis using all monthly data in 2000-2013, the OA-to-sulfate regression
slope is $m$=0.29 ($r^2$=0.25) from IMPROVE,  $m$=0.51 ($r^2$=0.43) from SEARCH, and $m$=1.87
($r^2$=0.57) from the default model, even though the default model well captures the magnitude,
trend, and monthly variability of observed sulfate. In summary, simulated total OA mass in the
standard GEOS-Chem model, dominated by IEPOX-SOA, has a steeper decreasing trend from
2000 to 2013 than the observations, and has a large MMV indicating strong dependence on
sulfate.

**3.2 What controls the modeled IEPOX-SOA variability?**
The strong dependence of IEPOX-SOA on sulfate is well-established by laboratory and field
work: wet sulfate particles provide the surface and volume of liquid media for IEPOX reactive
uptake (Budisulistiorini et al., 2017; Eddingsaas et al., 2010; Riva et al., 2016; Xu et al., 2015b,
2016), and serve as nucleophiles for nucleophilic addition to form organosulfates (Nguyen et al.,
2014; Surratt et al., 2007b). Sulfate ($SO_4^{2-}$), together with ammonium ($NH_4^+$), nitrate ($NO_3^-$) and
other ions, regulates proton ($H^+$) activity ($a_{H+}$) that can catalyze the ring-opening of epoxide





group leading to the formation of IEPOX-SOA (Gaston et al., 2014; Pye et al., 2013; Surratt et
al., 2007a). However, some recent studies suggest that IEPOX-SOA is not well correlated with
aerosol acidity estimated from thermodynamic models (Budisulistiorini et al., 2015; Lin et al.,
2013; Xu et al., 2015b), although the lack of direct measurements of aerosol acidity may be a
limitation. We use the GEOS-Chem model here to examine the simulated IEPOX-SOA
dependence on sulfate, aerosol acidity, and emissions of isoprene which produce IEPOX at high
yields under low-$NO_x$ conditions (Paulot et al., 2009). We do not treat aerosol water as an
independent driver because the dilution effect of aerosol water is implicitly considered in the
inorganic sulfate-ammonium-nitrate aerosol volume and acidity calculation, and studies have
shown that particle water is not a limiting factor unless the particle is purely dry (Nguyen et al.,
2014; Riva et al., 2016; Xu et al., 2015b) which is rare in summertime in the southeast US.

We find that the large MMV of OA in the model is mainly driven by sulfate concentrations and
aerosol acidity (Figure 3). Prior to 2008, IEPOX-SOA production is largely enhanced by
abundant sulfate (Gaston et al., 2014). Due to this high level of sulfate (about >4 µg/m$^3$), the
modeled aerosol acidity becomes particularly sensitive to variations in $NH_3$ emissions. For the
default model setup, we use monthly anthropogenic emissions from the EPA's National
Emissions Inventory 2011 (NEI11v1) in the US and adjust to each year from 2000 to 2013 using
national annual scaling factors (Travis et al., 2016), which suggests no significant long-term
trend of $NH_3$ emissions from 2000 to 2013 (Figure S3). The $NH_3$ emissions in August are about
25% lower than in June and July (Figure S3). As a result, the $a_{H+}$ in August is up to 3 times
higher than June, leading to high production of IEPOX-SOA in August. Both sulfate and aerosol
acidity appear to be the dominant contributors to MMV of OA during this period. After 2008,



IEPOX-SOA formation is substantially suppressed, due to small $SO_2$ emissions and low modeled
aerosol acidity $a_{H+}$ with small monthly variability. Isoprene emissions also contribute to the
month-to-month and interannual OA variability in the model.

The multivariate linear regression analysis of IEPOX-SOA quantitatively determines the relative
importance of its three drivers in the model. Using all monthly data in 2000-2013, the
standardized regression coefficients ($\beta$) associated with $a_{H+}$, sulfate aerosol concentration and
isoprene emission are $\beta$=0.50 ($r^2$=0.71), $\beta$=0.39 ($r^2$=0.64) and $\beta$=0.34 ($r^2$=0.18), respectively,
suggesting that aerosol acidity is the dominant controlling factor. The three variables together
explain 88% of the variability of IEPOX-SOA. Their relative importance changes over time
(Table S1). Aerosol acidity strongly correlates with IEPOX-SOA in 2005-2008 ($\beta$=0.57,
$r^2$=0.82) but its role becomes much weaker after 2008 ($\beta$=0.27, $r^2$=0.56). Sulfate aerosol is
always the first or second most important driver, especially in 2000-2004 ($\beta$=0.46, $r^2$=0.76).
Isoprene emission contributes to the overall interannual variability, for example leading to the
relatively low IEPOX-SOA in 2003-2004 and the peak in 2011 (Figure 3).

**3.3 Narrowing the gap between model and observation**
**3.3.1 Coating**
Several reasons may lead to the large monthly variations of the modeled OA. The modeled
IEPOX-SOA shows a much stronger sensitivity to aerosol acidity than suggested by field
observations, which found weak or no correlation between observed IEPOX-SOA and derived
aerosol acidity (Budisulistiorini et al., 2015; Lin et al., 2013; Worton et al., 2013; Xu et al.,
2015b). Lack of consideration of organic coating effect may provide one possible explanation. In



the real atmosphere, inorganic aerosol is generally internally mixed with other organics (Anttila
et al., 2006; Murphy et al., 2006). The presence of an organic coating may alter the solubility and
diffusion properties at the surface of inorganic particles and diminish further uptake of IEPOX.
By implementing a linear coating effect for the IEPOX uptake, both the magnitude of $\gamma_{IEPOX}$ and
its sensitivity to acidity have been reduced. Figure 4A shows a schematic illustrating the
dependence of the $\gamma_{IEPOX}$ coating effect on acidity $a_{H+}$ and organic mass fraction ($\chi_{org}$). The
original $\gamma_{IEPOX}$ without coating is represented at $\chi_{org}$=0. The orange line in Figure 4A shows the
approximate position of JJA-averaged acidity and organic mass fraction in the model simulation
with coating effect (simulation name: 'CT'). Adding a coating reduces $\gamma_{IEPOX}$ by almost half, but
the impact on the total reactive uptake rate of IEPOX is partially compensated by the
corresponding increase in particle surface area. The sensitivity of $\gamma_{IEPOX}$ to acidity has also been
reduced especially during the early 2000s (Figure 4A). The CT simulation reduces the southeast
US JJA-averaged IEPOX-SOA concentrations by 0.3~1.8 μg/m$^3$ (Figure 4C).

**3.3.2 NH$_3$ emissions and aerosol acidity**
Second, recent studies present contradictory results and explanations on the long-term trend of
aerosol acidity in the southeast US (Pye et al., 2019b; Silvern et al., 2017; Weber et al., 2016). In
this study, we show that the decreasing trend of aerosol acidity from the standard GEOS-Chem
model is mainly caused by high acidity in August before 2008, which corresponds to insufficient
NH$_3$ emissions in high sulfate environments. The NEI11v1 inventory is used in the default
configuration, in which NH$_3$ emissions in June and July are 30% higher than in August (Figure
S3), but not all NH$_3$ emission inventories agree with such pattern (Paulot et al., 2014). We did a
sensitivity test replacing the default US NH$_3$ emissions from NEI11v1 by a new NH$_3$ emission
product derived from CrIS satellite observations, which has higher emissions and smaller MMV
among June, July and August (Figure S3). In the simulation with updated $NH_3$ emissions in the
US from 2000 to 2013 ('CT_newNH$_3$'), the resulting simulated aerosol acidity is substantially
changed in 2000-2008 (Figure 4B). The high acidity ($a_{H+}$=0.55~0.9 mol/L) in August has been
reduced to around 0.2 mol/L and is much closer to June and July values (Figure 3B). The results
suggest that the fine particles in the southeast US are within a regime where the acidity ($a_{H+}$ in
units of mol/L) is sensitive to $NH_3$ emissions relative to sulfate concentration, though
corresponding pH changes are small (pH within 0.5~1.5, Figure S3). Small changes in $NH_3$ may
lead to large changes in $a_{H+}$ especially when sulfate concentrations are high, resulting in high
month-to-month variability of the IEPOX uptake. After updating the $NH_3$ emissions using the
satellite-based estimates, the model simulates a much more stable trend in aerosol acidity from
2000 to 2013 (Figure 4B), consistent with recent thermodynamic modeling studies that suggested
steady aerosol acidity despite large reductions in observed sulfate (Pye et al., 2019b; Weber et
al., 2016).

Due to the high uncertainty associated with the derived $NH_3$ emission product and acidity
calculation (Guo et al., 2015, 2018; Silvern et al., 2017; Song et al., 2018; Tao and Murphy,
2019), we conducted another simulation 'CT_H01' that fix $a_{H+}$ level at 0.1 mol/L when
calculating IEPOX uptake rate, corresponding to the observed $a_{H+}$ value during the 2013 SOAS
campaign (Weber et al., 2016). The two simulations, CT_newNH$_3$ and CT_H01, yield similar
long-term trends of IEPOX-SOA in the southeast US (Figure S4). For the SOAS2013 campaign
at Centerville, AL from 06/01/2013 to 07/15/2013, the CT_H01 scheme simulates an average
IEPOX-SOA concentration of 0.74 μg/m$^3$, similar to 0.81 μg/m$^3$ in the default model, and agrees





well with the two independent Aerosol Mass Spectrometer measurements (0.97 μg/m$^3$ from
obs_GT and 0.68 μg/m$^3$ from obs_CU, see daily time series in Figure S5). The CT_newNH$_3$
scheme simulates an average IEPOX-SOA concentration of 0.34 μg/m$^3$, lower than the
observation and the other models by a factor of >2, due to both the coating effect and small
aerosol $a_{H+}$ values ($a_{H+}$<0.1mol/L, Figure 4B). In general, the fixed acidity in the CT_H01
simulation well captures the measured IEPOX-SOA from the SOAS2013 campaign (Figure S5),
and improves the modeled total OA mass relative to the observations: The modeled long-term
decreasing rate of JJA-average OA from 2000 to 2013 has been reduced from 4.9%/year to
3.2%/year, better compared to the IMPROVE (1.7%/year) and SEARCH (1.9%/year)
observations, but is still higher (Figure 4C). The modeled MMV of OA have also been greatly
reduced (Figure 4D).

**3.3.3 Relationships between OA and sulfate**
The formation of aqueous SOA explicitly depends on sulfate aerosol and aerosol acidity which is
also impacted by sulfate. The default model, in which a large fraction of simulated total OA mass
is from aqueous SOA (mostly IEPOX-SOA), shows a stronger dependence of total OA on sulfate
than the observations (Figure 5). The OA-to-sulfate regression slope calculated using monthly
OA and sulfate (averaged from all sites beforehand for each network) is $m$=1.87 for the default
simulation, much higher than $m$=0.29 from IMPROVE and $m$=0.51 from SEARCH. Such strong
dependence is clearly demonstrated by the MMV of IEPOX-SOA (Figure 2).  Adding the
coating effect and fixing $a_{H+}$=0.1 mol/L substantially reduces the MMV of IEPOX-SOA and the
simulated monthly OA-to-sulfate slope ($m$=1.02).



Despite the model improvement against the observations in terms of OA and IEPOX-SOA
magnitude and long-term relationship with sulfate, the CT_H01 scheme needs to be further
improved. The rate of OA decreases per year in CT_H01 is about 0.8 times higher than the long-
term observations, with modeled MMV still larger than observations in early 2000s (Figure 4D).
Recent studies (Riva et al., 2019) suggested that the IEPOX-SOA production per unit mass of
sulfate likely increases with decreasing sulfate due to changes in aerosol properties, such as
acidity, morphology, phase state and viscosity. Further modeling studies with separated IEPOX-
SOA species and detailed aerosol properties are needed to achieve a better mechanistic
understanding of the dependence of OA on inorganic aerosol.

## 414    4. Summary and Discussion

Significant reduction of $SO_2$ emissions, combined with monthly variations of sulfate and $NH_3$
emissions, provide a unique dataset to test the sensitivity of biogenic SOA formation to inorganic
species. Observations from two networks (IMPROVE and SEARCH) show a slowly decreasing
trend in total OA mass from 2000 to 2013 in the southeast US (-1.7%/year from IMPROVE and -
1.9%/year from SEARCH), in contrast to a much faster rate of sulfate reduction (-6.9%/year
from IMPROVE and -6.7%/year from SEARCH). The standard version of GEOS-Chem model
was able to reproduce the long-term trend of sulfate (-6.7%/year), but with a faster decrease of
OA (-4.9%/year) and larger interannual variability.

The MMV of total OA mass during summers provides a novel observational constraint on SOA
formation mechanism. Remarkably, we find little MMV of OA from all three surface networks
(IMPROVE, SEARCH and CSN) during summer months in 2000-2013, despite larger MMV in



sulfate and NH$_3$ emissions. This is in contrast to the standard version of the GEOS-Chem model,
which shows a much larger MMV of OA during 2000-2008. Large MMV of OA in the standard
model is mainly due to the high sensitivity of modeled IEPOX-SOA to sulfate and aerosol
acidity (and NH$_3$ emissions) when sulfate aerosol is abundant. The resulting strong correlation
between OA and sulfate also appears to be at odds with long-term observations (Figure 5).
Incorporating a coating effect for IEPOX uptake and fixing aerosol acidity, have together
improved the model performance in terms of OA trend, variability and the relationship between
OA and sulfate, though further improvement is needed.

There are many uncertainties associated with the calculation of IEPOX-SOA formation. In the
default scheme, the Henry's law constant for IEPOX uptake was tuned using measurements from
the SOAS2013 campaign and was found to be $1.7 \times 10^7$ M/atm, 10 times smaller than suggested
by Gaston et al. (2014) based on laboratory experiments. The default simulation agrees well with
surface IEPOX-SOA data from SOAS2013 and SEAC4RS 2013 aircraft campaigns (Marais et
al., 2016) but overestimates OA magnitude and MMV against long-term observations from
IMPROVE and SEARCH. The CT_newNH$_3$ simulation reproduces the long-term OA trend but
underestimates IEPOX-SOA by a factor of 2 against SOAS 2013. The coating effect may be
stronger than used here, as Gaston et al. (2014) used a low viscosity organic material in the
experiments. The NH$_3$ emissions (which are critical for the calculation of aerosol acidity) are
highly uncertain (Dammers et al., 2019), and the acidity calculation is further complicated by
non-volatile cations (Guo et al., 2018) and meteorological conditions (Guo et al., 2015; Tao and
Murphy, 2019). Uncertainties are also associated with the volatility of IEPOX-SOA. Some
studies suggested a large fraction of IEPOX-SOA compounds (e.g. 2-methyltetrol) are semi-



volatile and can re-evaporate back into gas-phase (Ambro et al., 2019; Isaacman-VanWertz et
al., 2016), while other studies suggest IEPOX-SOA products are mostly nonvolatile or low
volatility (Hu et al., 2016; Lopez-Hilfiker et al., 2016). As multiple parameters may be tuned in
the model to fit observations, further laboratory, field and modeling studies are needed to
integrate Henry's law constant, IEPOX-SOA yields, volatility, coating effect and acidity
dependence for a better mechanistic understanding. The CT_H01 scheme lacks mechanical
representation of detailed aerosol properties like phase state, acidity, viscosity and morphology,
but reasonably captures both the OA and IEPOX-SOA magnitude (compared to both the three
filter measurement networks and the SOAS2013 campaign), long-term variability and
relationship with sulfate (Figure 4, 5, S5), therefore may serve as a simplified representation for
climate models. For all kinds of models, long-term filter-based measurements, especially
intraseasonal MMV, are important observational constraints that should be considered in model
development.

Even with our improved model, the rate of OA decrease per year is still 0.8 times higher the
long-term observations, and still shows a higher MMV than observations particularly in early
2000s (Figure 4D). Such discrepancies may suggest a more important role of SOA pathways that
are less dependent on inorganic aerosol, such as the gas-aerosol partitioning. Despite a large
MMV in IEPOX-SOA, Xu et al. (2015a) finds the less-oxidized oxygenated OA (LO-OOA, an
indicator for freshly-formed monoterpene SOA) and the more-oxidized oxygenated OA (MO-
OOA, also likely from biogenic sources) with little MMV in summer months, and they
contribute to more than 50% of total OA mass in the southeast US (Xu et al., 2018). The
important role of monoterpenes SOA is also confirmed by molecular level characterization of



organic aerosols (Zhang et al., 2018). Other pathways may contribute to SOA to some extent and
may add to the predicted SOA formed by partitioning, including biogenic SOA from auto-
oxidation (Bianchi et al., 2019; Pye et al., 2019a), in-cloud SOA formation that may be less
dependent on acidity than aqueous SOA (Tsui et al., 2019), a small but underestimated
contribution of anthropogenic SOA (Schroder et al., 2018; Shah et al., 2019), and other possible
mechanisms (Schwantes et al., 2019). Further quantifying the relative importance of the different
pathways will allow a more accurate quantification of the anthropogenic influence on biogenic
SOA and the associated radiative forcing.




**Data availability**
The observational datasets from long-term filter measurement networks IMPROVE and CSN are available at http://views.cira.colostate.edu/fed/QueryWizard/Default.aspx. The SEARCH observational datasets are available by contacting E. Edgerton. The model code and modeling results are available by contacting Y. Zheng and J. Mao.

**Acknowledgement**
YZ and JM acknowledge funding from NOAA NA18OAR4310114. HC and DKJ recognize support from NASA 80NSSC18K0689. WH and JLJ acknowledge funding from NSF AGS-1822664. EAM acknowledges funding from NERC/EPSRC (award number EP/R513465/1). The authors acknowledge the Electric Power Research Institute (EPRI) and Southern Company for support of the SEARCH network and Atmospheric Research & Analysis. IMPROVE and CSN data are accessed from the Federal Land Manger Environmental Database. YZ thanks helpful discussions with Arlene M. Fiore, Róisín Commane and V. Faye McNeill.

**Author Contributions:**
Y.Z. and J.M. designed the research, performed the simulations and conducted the analysis. J.A.T. and E.M.M. provided guidance on aerosol coating parameterization. H.C. and D.K.H. provided the CrIS-derived $NH_3$ emission. N.L.N, W.H. and J.L.J. provided data from the SOAS2013 field campaign. E.E. provided data from the SEARCH network. Y.Z. wrote the paper with all coauthors providing input.

**Competing interests**
The authors declare no competing interests.

**Additional information**
Correspondence and material requests should be addressed to Y. Zheng and J. Mao.





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



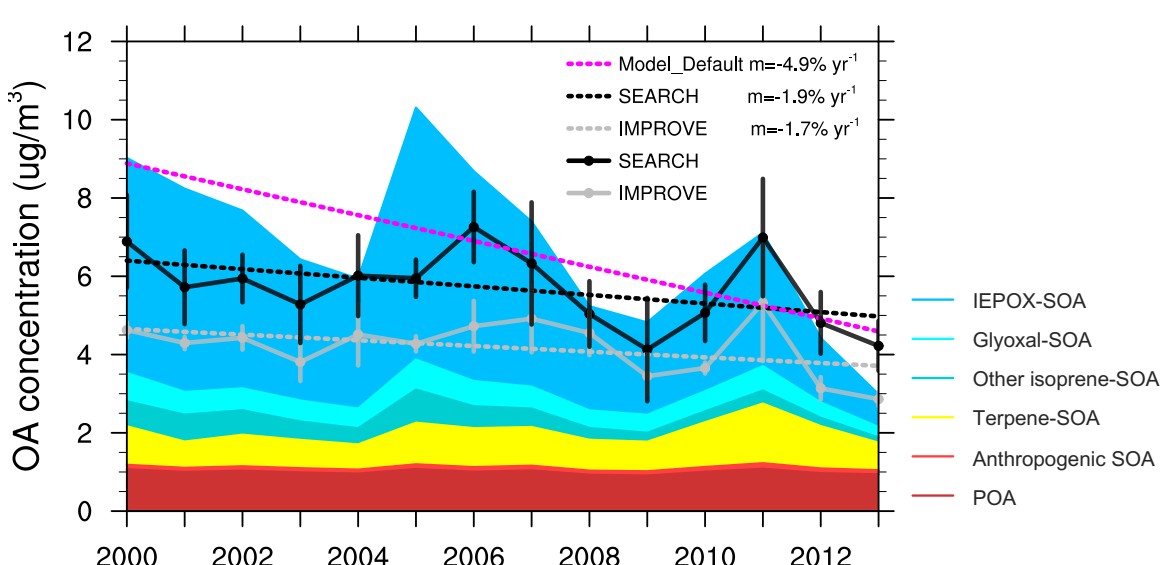

**Figure 1.** Comparison of June-July-August averaged surface OA concentration (µg/m³) over the southeast US between the default model and the observation from IMPROVE and SEARCH network. Colored shades represent different components of modeled OA. IEPOX-, glyoxal-, and other isoprene-SOA are from aqueous uptake of isoprene oxidation products. Terpene- and anthropogenic SOA are dry SOA calculated using volatility-basis-set.





**Figure 2. (A)** Monthly surface OA and sulfate ($SO_4^{2-}$) concentration (µg/m³) averaged over the southeast US from IMPROVE, SEARCH and the default model. **(B)** Monthly surface concentrations of IEPOX-SOA and the sum of POA and dry SOA from the default model, and IEPOX-SOA from the CT_H01 simulation.

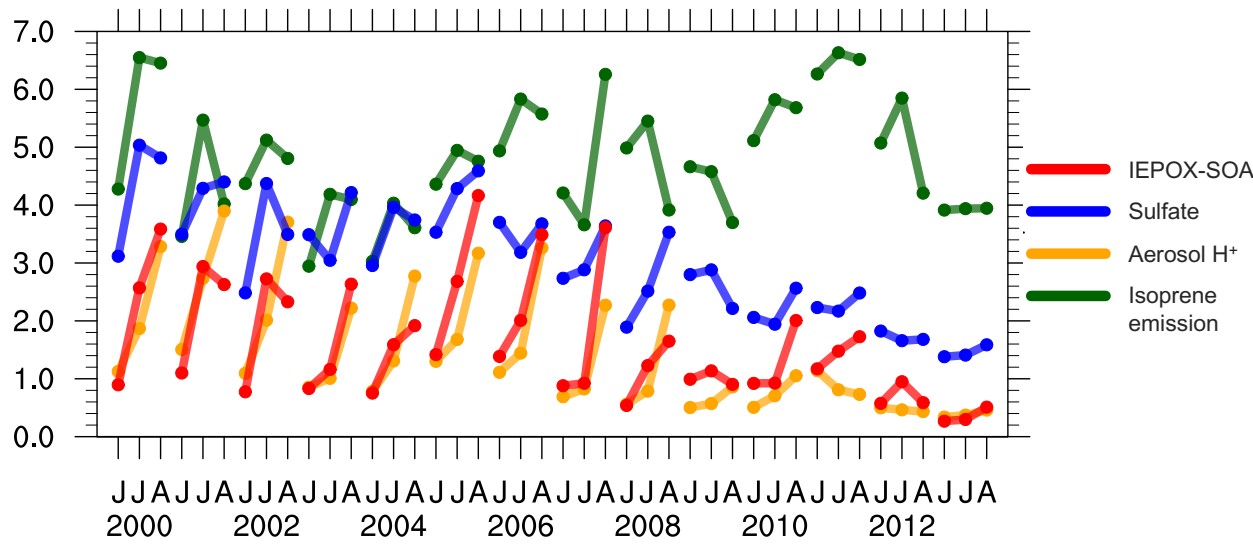

**Figure 3.** Standardized monthly surface IEPOX-SOA concentration, sulfate concentration, aerosol $H^+$ activity and isoprene emission from the default model. All variables are averaged over the southeast US, and have been divided by 1 standard deviations, therefore are unitless.



**Figure 4**. **(A)** Schematic diagram of IEPOX reactive uptake coefficient ($\gamma_{IEPOX}$). Colored lines indicate the position of JJA-averaged organic mass fraction and aerosol H$^+$ activity in 2000-2013 from the 'Default', 'CT' and 'CT_H01' simulations. **(B)** Simulated aerosol acidity (mol/L) from the default, 'CT_newNH$_3$' and 'CT_H01' simulations. **(C)** JJA-averaged surface OA (µg/m$^3$) from IMPROVE, SEARCH and all model simulations. **(D)** Standard deviation of OA (µg/m$^3$) between June, July and August from IMPROVE, SEARCH and all model simulations. All results are averaged over the southeast US.

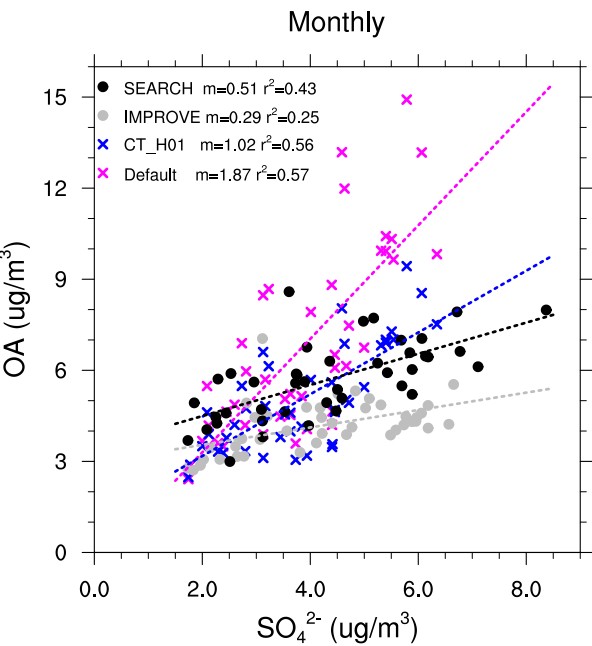

**Figure 5.** Relationships between monthly OA and sulfate concentrations ($\mu g/m^3$). Each dot represents monthly data averaged from all sites from each network within the southeast US.