# Peer review of "Long-term observational constraints of organic aerosol dependence on inorganic species in the southeast US"

_Atmospheric Chemistry and Physics, 2020_

## Referee Comment (RC1) · Anonymous Referee #1 · 23 Jul 2020

This work looks at the modeled vs. observed trends in sulfate and organic aerosol in the southeast US and finds that the model OA is overly dependent on sulfate, due to the impact of sulfate on acid-catalyzed reactive uptake of epoxydiols. The authors use the month-to-month variability in OA during summers to provide a new constraint on SOA formation mechanisms. Overall, this paper is well-written and within the scope of ACP and should be published after minor revisions.

Major comments.
1.  Page 8, line 151 – The statement "The $2°x2.5°$ simulations are adequate when modeling continental boundary layer chemistry (Yu et al., 2016)" is true for ozone and $NO_x$, but not for isoprene oxidation pathways. Yu et al. showed a large shift from high-NOx to low-NOx oxidation (Fig. 3), which would impact IEPOX production and SOA from isoprene. Please address this limitation of your work, and how an overestimate in the high $NO_x$ pathway would impact your results. Possibly a small sensitivity study is warranted to address this issue. Could it be that the model actually would have even more IEPOX SOA at high resolution and the problem would be worse?
2.  Is there a reason you didn't use observations past 2013?
3.  Could you comment on how the lack of nonvolatile cations in ISORROPIA might impact your calculated aerosol acidity and generally effect your results?
    (https://pdfs.semanticscholar.org/f642/9d2c07179b7624c795aed5bf37b20aa0e2a8.pdf)

Minor comments.
4.  Page 1, line 34 – The statement "Biogenic SOA may account for 60-100% of OA" is confusing, if you are going to call it biogenic SOA, but then discuss anthropogenic influence, maybe call it something else, like biogenic-derived SOA or similar.
    a.  Also, Kim et al., 2015 say that biogenic isoprene and monoterpenes account for 60 % of OA, where do you get the 100% value?
5.  Page 5, line 70 – What about the role of NOx (and nitrate) in regulating aerosol water?
6.  Figure S1 – it is implied that the CSN sites are in Figure S1, which only shows SEARCH and IMPROVE.
7.  Page 7, line 126 – Where does the baseline model agricultural NH3 come from?
8.  Page 7 line 126 – Do you mean you scale the emissions from 2011 backwards to 2000 and forwards to 2013 using the annual emission totals from EPA? What do you mean by "mapped over $0.1°x0.1°$"
9.  Page 8, line 134 – The GEOS-Chem complex scheme is also described in great detail by Pai et al., 2020. "An evaluation of global organic aerosol schemes using airborne observations"
10. Page 8 line, 148. It is confusing to dive in and describe sensitivity studies without showing us the problem in Figure 1. I don't think it would be out of place to say that the default model, shown in Figure 1, overestimates the trend in organic aerosol largely due to … and we run 4 sensitivity studies to address this discrepancy…
11. Page 9, line 154 – It would be helpful to tell the reader that these sensitivity simulations will be described in Sections 2.2.2 etc..
12. Page 9, line 163 – do you mean at "the" or "from the" SOAS2013 campaign (citation?)
13. Do the authors expect that not including non-volatile cations in ISOROPPIA could impact their findings?
14. Page 10, line 178 – Can you provide some evidence/citations for us that there might be such a coating effect? Or is the ethylene glycol supposed to be representative enough of real organic aerosol coating ammonium bisulfate? Please just clarify a little more here.
15. Page 10, line 189 - Please elaborate a little more on the findings of Jo et al., 2019 since this seems important.
16. In Section 2.2.3 – How do your emissions results compare to the NH3 emissions derived from wet deposition in Paulot et al., 2014? *Paulot F., Jacob, D.J., Pinder R.W., Bash J.O., Travis, K., Henze D.K., Ammonia emissions in the United States, Europe, and China derived by high-resolution inversion of ammonium wet deposition data: Interpretation with a new agricultural emissions inventory (MASAGE_NH3), J. Geophys. Res., 119, 4,343-4,364, 2014.*
17. Page 13, line 241 – Do the sulfate trends refer to Figure 2? If so, please reference it here.

18. Page 11, line 199 – Is this due to improved farming practices as output increases?  Or some other explanation?  I am surprised NH₃ is not increasing and would assume other readers might appreciate a little more detail.
19. Section 2.2.3 – It would help the reader if there was a plot of the bottom-up and top-down NH3 emissions.
20. Page 11, line 208 – What surface observations? Please clarify what you mean, are there separate independent NH3 surface observations you compare against?  Again, a plot would be useful here.  Same for the seasonal cycle.  These can be in the supplement according to the author's preference.
21. Page 11, line 217 – For clarity, please specific that these are "model" IEPOX-SOA, sulfate aerosol etc.
22. Page 13, line 244 – Why not sample the model at the locations of the sites?  Would that make a difference to your trend?
23. Figure 1 – can you explain the periods of enhanced OA around 2006 and 2011?
24. Figure 2 – It could be helpful to include the trends on this figure similar to Figure 1.
25. Page 15, line 302 – Figure 3 is difficult to interpret, please provide a little more description of what is shown here, particularly what is mean by "divided by 1 standard deviations."
26. Page 16, line 325 – Is isoprene interannual variability explained by temperature? Temperature is an important driver of aerosol partitioning, should it not be included in Table S1 and Figure 3? Does temperature explain any of the interannual variation in Figure 1?
27. Page 18, line 376 – I assume that the H+ value during 2013 SOAS was calculated with a thermodynamic model, although constrained by observations.  Please clarify.
28. Page 19, line 378 – Can you comment on why the model agrees much better with SOAS2013 observations and 2013 OA in Figure 1, and what the limitations are of testing your revised IEPOX uptake at the low end of the SO₂ emissions trend?
29. General comment – why not consider trends in ammonium?  Particularly to help support your improved NH3 emissions scenario?
30. Page 22 , line 467 – By gas-aerosol partitioning, do you mean reversible uptake of isoprene oxidation products?  Could you be more specific on this?
    a. Generally, the final paragraph that goes through potential reasons for the remaining model discrepancies is very useful but needs additional explanation. The sentence starting "Despite a large MMV in IEPOX-SOA is confusing, please rephrase to more clearly state your meaning.
    b. Is monoterpene SOA included in the GEOS-Chem simulations?  If so, is the lack of MMV captured by the model?  I am generally just confused by the discussion here. Are you trying to say that other SOA pathways, not in the model, might have less MMV and therefore improve model performance of the long-term trend?

---

## Referee Comment (RC2) · Anonymous Referee #2 · 4 Aug 2020

Summary: The author compared the long-term trend of organic aerosol and sulfate mass loading from field measurement with GEOS-Chem simulations in this manuscript, with the model showing a steeper decrease in the OA mass loading and larger month-to-month variability than the field data. The long-term trend of sulfate, on the other hand, was well captured by the model when comparing with the field data. By applying coating effect, constant aerosol acidity, and a different NH3 emission product, the modeling results match the field data better, suggesting further study is needed to address the weak dependence of OA formation and sulfate. The manuscript is overall clearly written, but may need to address the following aspects before publishing.

[Figure]

First, the author did a good job explaining the modeling results in Section 3, but it was not very clear at certain sections which modeling results agree with the field measurement and which do not. The author may need to improve clarity about the modeling-field data comparisons when describing the modeling results, especially in line 251 when the author discusses that the contribution of IEPOX-SOA to total OA mass decreases from 61% to 28% from the early 2000s to 2013. Does this modeling result agree with previous field measurements (such as the results from Xu. et al. 2015 and Budisulistiorini,et al. 2016)? Maybe the author can include a sentence or two to compare the modeling data with the field data.

The manuscript also concluded that coating can improve the modeling result because thinner coating may enhance the formation of IEPOX-SOA. There are a few papers that also measured/modelled the effects of SOA coating on the formation of IEPOX-SOA. For instance, Zhang et al. 2018, Jo et al., 2019, and the subsequent study by Schmedding et al. 2019 discussed the effects of pre-existing coating on the formation of IEPOX-SOA. Does the result in this manuscript using ethylene glycol agree with previous studies using authentic SOA?

In addition, Jo et al. 2019 shows that the uptake of IEPOX would increase with increasing coating for most of the situations using GEOS-Chem due to increasing surface tension, contrary to a decreasing uptake with coating effect in this manuscript. Could the author explain why the trends are different in these two studies?

The manuscript used a fixed acidity to reduce the modelled month-to-month variation of IEPOX-SOA so that the results match better with the field data. It seems this large month-to-month variation in modeling only appeared before 2008, while the month-to-month variation decreased significantly after 2008 even with the default modeling scenario. Can the author explain why there is such a large difference in the month-to-month variation before and after 2008 in the default modeling scenario? Was it due to difference in NH3 emission inventory before and after 2008 or other reasons? In addition, after updating the NH3 inventory with the new emission inventory from CrIS

satellite observation, the author stated in line 384-385 that this scenario (CT_newNH3) performed a bit worse than the fixed acidity scenario (CT_H01). I wonder whether that was due to coating effects not correctly represented by using ethylene glycol rather than the results from authentic coatings. Besides the paper mentioned above, Li et al. 2020 and Zhang et al. also used simplified equations that can estimate the phase state of a few IEPOX-SOA species that might be helpful in performing future modeling.

Minor comments: Why would there be a large increase of the default IEPOX-SOA during 2005-2008? The manuscript mentioned about higher correlation of IEPOX-SOA with acidity during this period of time. Was this abrupt increase of IEPOX-SOA caused by lower NH3 emissions between 2005-2008?

Line 171: There are multiple papers discussing about different Henry's law constants for IEPOX. The author did discuss in line 439 but probably should include other relevant papers, such as Woo et al., 2015, Budisulistiorini et al., 2016, Pye et al., 2017, and Zhang et al., 2018.

Line 477-478: One other potential mechanism I can think of is the non-linear feedback between sulfate and IEPOX-SOA production discussed in recent studies. For instance, Riva et al. 2019 and Zhang et al. both show that IEPOX-SOA fraction could and sulfate are nonlinear due to chemical reactions, acidity, and the coating effects of IEPOX-SOA are intertwined and nonlinear due to the formation of organosulfates.

References: Xu, L. et al. Effects of anthropogenic emissions on aerosol formation from isoprene and monoterpenes in the southeastern United States. Proc. Natl. Acad. Sci. USA 112, 37-42, doi:10.1073/pnas.1417609112 (2015).

Woo, J. L. & McNeill, V. F. simpleGAMMA v1.0 – a reduced model of secondary organic aerosol formation in the aqueous aerosol phase (aaSOA). Geosci. Model Dev. 8, 1821-1829, doi:10.5194/gmd-8-1821-2015 (2015).

Budisulistiorini, S. H. et al. Seasonal characterization of submicron aerosol chemi-

cal composition and organic aerosol sources in the southeastern United States: Atlanta, Georgia,and Look Rock, Tennessee. Atmos. Chem. Phys. 16, 5171-5189, doi:10.5194/acp-16-5171-2016 (2016).

Budisulistiorini, S. H. et al. Simulating Aqueous-Phase Isoprene-Epoxydiol (IEPOX) Secondary Organic Aerosol Production During the 2013 Southern Oxidant and Aerosol Study (SOAS). Environ. Sci. Technol. 51, 5026-5034, doi:10.1021/acs.est.6b05750 (2017). Pye, H. O. T. et al. On the implications of aerosol liquid water and phase separation for organic aerosol mass. Atmos. Chem. Phys. 17, 343-369, doi:10.5194/acp-17-343-2017 (2017).

Zhang, Y. et al. Effect of Aerosol-Phase State on Secondary Organic Aerosol Formation from the Reactive Uptake of Isoprene-Derived Epoxydiols (IEPOX). Environ. Sci. Technol. Lett. 5, 167-174, doi:10.1021/acs.estlett.8b00044 (2018).

Schmedding, R. et al. $\alpha$-Pinene-Derived organic coatings on acidic sulfate aerosol impacts secondary organic aerosol formation from isoprene in a box model. Atmos. Environ., 456-462, doi:https://doi.org/10.1016/j.atmosenv.2019.06.005 (2019).

Jo, D. S. et al. A simplified parameterization of isoprene-epoxydiol-derived secondary organic aerosol (IEPOX-SOA) for global chemistry and climate models: a case study with GEOS-Chem v11-02-rc. Geosci. Model Dev. 12, 2983-3000, doi:10.5194/gmd-12-2983-2019 (2019).

Zhang, Y. et al. The Cooling Rate- and Volatility-Dependent Glass-Forming Properties of Organic Aerosols Measured by Broadband Dielectric Spectroscopy. Environ. Sci. Technol. 53, 12366-12378, doi:10.1021/acs.est.9b03317 (2019).

Zhang, Y. et al. Joint Impacts of Acidity and Viscosity on the Formation of Secondary Organic Aerosol from Isoprene Epoxydiols (IEPOX) in Phase Separated Particles. ACS Earth and Space Chemistry 3, 2646-2658, doi:10.1021/acsearthspacechem.9b00209 (2019).

Li, Y., Day, D. A., Stark, H., Jimenez, J. L. & Shiraiwa, M. Predictions of the glass transition temperature and viscosity of organic aerosols from volatility distributions. Atmos. Chem. Phys. 20, 8103-8122, doi:10.5194/acp-20-8103-2020 (2020).
* * *

---

## Author Comment (AC2) · 16 Sep 2020

**Response to Reviewer #2**

We are grateful to the reviewer for the helpful comments and guidance that have led to important improvements of the original manuscript. Our point-by-point responses are listed below. Reviewer's comments are in black font, and authors' responses are in dark blue. Page and line numbers refer to the discussion paper acp-2020-575. The revised figures and Supplementary Information are attached in the end.

Summary: The author compared the long-term trend of organic aerosol and sulfate mass loading from field measurement with GEOS-Chem simulations in this manuscript, with the model showing a steeper decrease in the OA mass loading and larger monthto-month variability than the field data. The long-term trend of sulfate, on the other hand, was well captured by the model when comparing with the field data. By applying coating effect, constant aerosol acidity, and a different NH3 emission product, the modeling results match the field data better, suggesting further study is needed to address the weak dependence of OA formation and sulfate. The manuscript is overall clearly written, but may need to address the following aspects before publishing.

First, the author did a good job explaining the modeling results in Section 3, but it was not very clear at certain sections which modeling results agree with the field measurement and which do not. The author may need to improve clarity about the modeling-field data comparisons when describing the modeling results, especially in line 251 when the author discusses that the contribution of IEPOX-SOA to total OA mass decreases from 61% to 28% from the early 2000s to 2013. Does this modeling result agree with previous field measurements (such as the results from Xu. et al. 2015 and Budisulistiorini, et al. 2016)? Maybe the author can include a sentence or two to compare the modeling data with the field data.

Section 3.1 mainly compares modeling results with long-term surface filter measurements from IMPRAOVE and SEARCH networks, focusing on the long-term trend and month-to-month variability of OA (Figure 1 and 2). We also compare the modeling results with the SOAS2013 field campaign data later in Section 3.3.2 Line 379-385 and Figure S6.

Here we add a sentence about the contribution of IEPOX-SOA: "The contribution of IEPOX-SOA to total OA mass decreases from 61% in the early 2000s to 28% in 2013. The simulated IEPOX-SOA in 2013 compares well with previous field studies which suggested that IEPOX-SOA contributed to 18~40% in southeast US sites in summer 2013 (Budisulistiorini et al., 2016; Xu et al., 2015)."

The manuscript also concluded that coating can improve the modeling result because thinner coating may enhance the formation of IEPOX-SOA. There are a few papers that also measured/modelled the effects of SOA coating on the formation of IEPOX-SOA. For instance, Zhang et al. 2018, Jo et al., 2019, and the subsequent study by Schmedding et al. 2019 discussed the effects of pre-existing coating on the formation of IEPOX-SOA. Does the result in this manuscript using ethylene glycol agree with previous studies using authentic SOA? In addition, Jo et al. 2019 shows that the uptake of IEPOX would increase with increasing coating for most of the situations using GEOS-Chem due to increasing surface tension, contrary to a decreasing

uptake with coating effect in this manuscript. Could the author explain why the trends are different in these two studies?

Compared to the default GEOS-Chem with no coating effect at all, adding a coating effect reduces (not enhances) the production of IEPOX-SOA in the summertime southeast US (Figure 4C, orange line as compared to magenta line). Jo et al. applied the parameterizations from Zhang et al., 2018 (using monoterpene-SOA as a coating material) and is different from what we used here. In Gaston et al. (2014) which we applied in this study, ethylene glycol is a relatively low viscosity material, but the simplified linear function fitted using RH=50% conditions may mimic a strong coating effect, because it does not consider the reduced viscosity and weaker coating effect at higher RH conditions, and it assumes no IEPOX uptake when the fraction of organics is higher than 70%. As a result, the added aerosol surface and particle radius does not overweigh the impact of reduced uptake coefficient when considering coating.

In Section 2.2.2 Line 180 we add: "In the real atmosphere when inorganic cores are coated with more viscous SOA (Zhang et al., 2018b), coating effect may be stronger because ethylene glycol is a low viscosity material. However, this simplified linear function does not consider the decreased viscosity and reduced coating effect at higher RH conditions (which is common in summertime southeast US) (Gaston et al., 2014; Zhang et al., 2018b), and prevents further IEPOX uptake when the mass fraction of OA ( $\chi_{org}$ ) is larger than 0.7, therefore this linear function may mimic a strong coating effect even though ethylene glycol is less viscous than real atmospheric SOA. The uncertainties need to be addressed in further studies with a more realistic coating parameterization (Li et al., 2020; Schmedding et al., 2019; Zhang et al., 2019b). We assume all OA is coated outside the inorganic aerosol core when calculating the IEPOX reactive uptake. The default GEOS-Chem with no organic coating calculates surface area of inorganic aerosol. By adding the coating effect, the increased particle radius  $R_p$  and surface area  $S_a$  of the mixed particle will partially offset (but does not overweigh) the impact of reduced reaction probability  $\gamma_{IEPOX\_modified}$ ."

The manuscript used a fixed acidity to reduce the modelled month-to-month variation of IEPOX-SOA so that the results match better with the field data. It seems this large month-to-month variation in modeling only appeared before 2008, while the month-to-month variation decreased significantly after 2008 even with the default modeling scenario. Can the author explain why there is such a large difference in the month-to-month variation before and after 2008 in the default modeling scenario? Was it due to difference in NH3 emission inventory before and after 2008 or other reasons?

We explained the large MMV before 2008 in Section 3.2. The main idea is that when sulfate is high before 2008, the relatively low  $NH_3$  emission in August is insufficient relative to sulfate and leads to a much higher aerosol H+ (than in June and July), which then leads to high IEPOX-SOA. After 2008 when sulfate becomes small, aerosol H+ becomes less sensitive to the supply of  $NH_3$ , so even there are still monthly differences in  $NH_3$  emissions, the MMV of IEPOX-SOA is smaller.

In Section 2.2.2 At Line 301-303 we have the explanation: "Prior to 2008, IEPOX-SOA production is largely enhanced by abundant sulfate. Due to this high level of sulfate (about >4

 $\mu$ g/m3), the modeled aerosol acidity becomes particularly sensitive to variations in NH3 emissions. ... The NH3 emissions in August are about 25% lower than in June and July (Figure S4). As a result, in August before 2008, the aerosol NH4+/SO42- ratio is smaller (Figure S4) and aH+ is up to 3 times higher than June (Figure 4B), leading to high production of IEPOX-SOA in August. Both sulfate and aerosol acidity appear to be the dominant contributors to MMV of OA during this period. After 2008, IEPOX-SOA formation is substantially suppressed, due to small SO2 emissions and low modeled aerosol acidity aH+ with small monthly variability."

In addition, after updating the NH3 inventory with the new emission inventory from CrIS satellite observation, the author stated in line 384-385 that this scenario (CT\_newNH3) performed a bit worse than the fixed acidity scenario (CT\_H01). I wonder whether that was due to coating effects not correctly represented by using ethylene glycol rather than the results from authentic coatings. Besides the paper mentioned above, Li et al. 2020 and Zhang et al. also used simplified equations that can estimate the phase state of a few IEPOX-SOA species that might be helpful in performing future modeling.

Thank you for the suggestions. The CT\_newNH3 simulation may be influenced by both the simplified coating scheme as well as uncertainties associated with NH3 emissions. In the text we have:

"The CT\_newNH3 scheme simulates an average IEPOX-SOA concentration of 0.34  $\mu$ g/m3, lower than the observation and the other models by a factor of >2, due to both the simplified coating effect and small aerosol  $a_{H^+}$  values ( $a_{H^+}$ <0.1mol/L, Figure 4B)." The Li et al. and Zhang et al. papers have been added to the model description Section 2.2.2 (see response to the second comment).

Minor comments: Why would there be a large increase of the default IEPOX-SOA during 2005-2008? The manuscript mentioned about higher correlation of IEPOX-SOA with acidity during this period of time. Was this abrupt increase of IEPOX-SOA caused by lower NH3 emissions between 2005-2008?

The high IEPOX-SOA in 2005-2007 is a result of both high sulfate, high aerosol acidity in August and high isoprene emissions.

In Section 3.2 at Line 314, we add: "The high IEPOX-SOA in 2000-2001 and 2005-2007 is a result of high sulfate aerosol, high aerosol acidity due to low NH3 supply relative to high sulfate, and high isoprene emissions during these periods (Figure 3, Figure 4B)."

Line 171: There are multiple papers discussing about different Henry's law constants for IEPOX. The author did discuss in line 439 but probably should include other relevant papers, such as Woo et al., 2015, Budisulistiorini et al., 2016, Pye et al., 2017, and Zhang et al., 2018.

Thank you for the suggestions. We add at Line 439:

"the Henry's law constant for IEPOX uptake was tuned using measurements from the SOAS2013 campaign and was found to be  $1.7 \times 10^7$  M/atm, 10 times smaller than suggested by

Gaston et al. (2014) based on laboratory experiments and about half of the suggested value (3×107 M/atm) in some other studies (Budisulistiorini et al., 2017; Nguyen et al., 2014; Pye et al., 2017; Woo and McNeill, 2015; Zhang et al., 2018b)."

Line 477-478: One other potential mechanism I can think of is the non-linear feedback between sulfate and IEPOX-SOA production discussed in recent studies. For instance, Riva et al. 2019 and Zhang et al. both show that IEPOX-SOA fraction could and sulfate are nonlinear due to chemical reactions, acidity, and the coating effects of IEPOX-SOA are intertwined and nonlinear due to the formation of organosulfates.

Thank you for the suggestions. We mentioned the non-linearity at Line 408-410, and the Zhang et al. paper has been added as a reference now:

"Recent studies (Riva et al., 2019) suggested that the IEPOX-SOA production per unit mass of sulfate likely increases with decreasing sulfate due to changes in aerosol properties, such as acidity, morphology, phase state and viscosity, as well as formation of organosulfates, suggesting non-linearity between IEPOX-SOA and sulfate (Riva et al., 2019; Zhang et al., 2019)."

**References:**

Budisulistiorini, S. H. et al. Seasonal characterization of submicron aerosol composition and organic aerosol sources in the southeastern United States: Atlanta, Georgia, and Look Rock, Tennessee. Atmos. Chem. Phys. 16, 5171-5189, doi:10.5194/acp-16-5171-2016 (2016).

Budisulistiorini, S. H. et al. Simulating Aqueous-Phase Isoprene-Epoxydiol (IEPOX) Secondary Organic Aerosol Production During the 2013 Southern Oxidant and Aerosol Study (SOAS). Environ. Sci. Technol. 51, 5026-5034, doi:10.1021/acs.est.6b05750 (2017).

Li, Y., Day, D. A., Stark, H., Jimenez, J. L. & Shiraiwa, M. Predictions of the glass transition temperature and viscosity of organic aerosols from volatility distributions. Atmos. Chem. Phys. 20, 8103-8122, doi:10.5194/acp-20-8103-2020 (2020).

Pye, H. O. T. et al. On the implications of aerosol liquid water and phase separation for organic aerosol mass. Atmos. Chem. Phys. 17, 343-369, doi:10.5194/acp17-343-2017 (2017).

Schmedding, R. et al.  $\alpha$ -Pinene-Derived organic coatings on acidic sulfate aerosol impacts secondary organic aerosol formation from isoprene in a box model. Atmos. Environ., 456-462, doi:https://doi.org/10.1016/j.atmosenv.2019.06.005 (2019).

Jo, D. S. et al. A simplified parameterization of isoprene-epoxydiol-derived secondary organic aerosol (IEPOX-SOA) for global chemistry and climate models: a case study with GEOS-Chem v11-02-rc. Geosci. Model Dev. 12, 2983-3000, doi:10.5194/gmd12-2983-2019 (2019).

Woo, J. L. & McNeill, V. F. simpleGAMMA v1.0 – a reduced model of secondary organic aerosol formation in the aqueous aerosol phase (aaSOA). Geosci. Model Dev. 8, 1821-1829, doi:10.5194/gmd-8-1821-2015 (2015).

Xu, L. et al. Effects of anthropogenic emissions on aerosol formation from isoprene and monoterpenes in the southeastern United States. Proc. Natl. Acad. Sci. USA 112, 37-42, doi:10.1073/pnas.1417609112 (2015).

Zhang, Y. et al. Effect of Aerosol-Phase State on Secondary Organic Aerosol Formation from the Reactive Uptake of Isoprene-Derived Epoxydiols (IEPOX). Environ. Sci. Technol. Lett. 5, 167-174, doi:10.1021/acs.estlett.8b00044 (2018).

Zhang, Y. et al. Joint Impacts of Acidity and Viscosity on the Formation of Secondary Organic Aerosol from Isoprene Epoxydiols (IEPOX) in Phase Separated Particles. ACS Earth and Space Chemistry 3, 2646-2658, doi:10.1021/acsearthspacechem.9b00209 (2019).

Zhang, Y. et al. The Cooling Rate- and Volatility-Dependent Glass-Forming Properties of Organic Aerosols Measured by Broadband Dielectric Spectroscopy. Environ. Sci. Technol. 53, 12366-12378, doi:10.1021/acs.est.9b03317 (2019).

**Figure 1.** Comparison of June-July-August averaged surface OA concentration ( $\mu g/m^3$ ) over the southeast US between the default model and the observation from IMPROVE and SEARCH network. Colored shades represent different components of modeled OA. IEPOX-, glyoxal-, and other isoprene-SOA are from aqueous uptake of isoprene oxidation products. Terpene- and anthropogenic SOA are dry SOA calculated using volatility-basis-set.

---

## Author Response (AR1)

**Response to Reviewer #1**

We are grateful to the reviewer for the helpful comments and guidance that have led to important improvements of the original manuscript. Our point-by-point responses are listed below. Reviewer's comments are in black font, and authors' responses are in dark blue. Page and line numbers refer to the discussion paper acp-2020-575. The revised figures and Supplementary Information are attached in the end.

This work looks at the modeled vs. observed trends in sulfate and organic aerosol in the southeast US and finds that the model OA is overly dependent on sulfate, due to the impact of sulfate on acid-catalyzed reactive uptake of epoxydiols. The authors use the month-to-month variability in OA during summers to provide a new constraint on SOA formation mechanisms. Overall, this paper is well-written and within the scope of ACP and should be published after minor revisions.

**Major comments.**

1. Page 8, line 151 - The statement "The 2x2.5 simulations are adequate when modeling continental boundary layer chemistry (Yu et al., 2016)" is true for ozone and NOx, but not for isoprene oxidation pathways. Yu et al. showed a large shift from high-NOx to low-NOx oxidation (Fig. 3), which would impact IEPOX production and SOA from isoprene. Please address this limitation of your work, and how an overestimate in the high NOx pathway would impact your results. Possibly a small sensitivity study is warranted to address this issue. Could it be that the model actually would have even more IEPOX SOA at high resolution and the problem would be worse?

Figure 3 in Yu et al. actually showed a shift from low-NOx to high-NOx regime when switching from a coarse grid to a fine grid, which may help alleviate the issue of IEPOX-SOA overestimation in this paper (because IEPOX is a product from the low-NOx pathway of isoprene oxidation). Due to the memory limitation of our local supercomputer system, we were unable to conduct a nested simulation at fine resolution using the complexSOA scheme.

We remove the sentence citing Yu et al. (2016) in the model description in Section 2.2.1 and add the following discussion in Section 4 Line 460: "The CT\_H01 scheme ... may serve as a simplified representation for climate models. **Simulations in this study are conducted at a horizontal resolution of 2°×2.5°**, which is comparable to most global climate models. However, as shown by Yu et al. (2016), from coarse to fine horizontal resolution, there will be a shift from low-NOx to high-NOx pathway for isoprene oxidation. Therefore, using a fine resolution may reduce the production of IEPOX and IEPOX-SOA, which needs further investigation. For all kinds of models, long-term filter-based measurements, especially intraseasonal MMV, are important observational constraints that should be considered in model development."

2. Is there a reason you didn't use observations past 2013?

We use the 2000-2013 observations to compare with our model results because we use the NEI2011 emission inventory in our model simulations, which does not provide annual emission

scaling factors after year 2013. We may update to NEI2014 which includes information for emissions after 2013 in future studies. As the main goal of this study is to show and explain the discrepancies between model and observations in early 2000s, we think using the current NEI2011 emission inventory is sufficient for this study.

3. Could you comment on how the lack of nonvolatile cations in ISORROPIA might impact your calculated aerosol acidity and generally effect your results? (https://pdfs.semanticscholar.org/f642/9d2c07179b7624c795aed5bf37b20aa0e2a8.pdf)

Theoretically, the lack of nonvolatile cations (NVCs) in ISORRPIA may lead to a higher aerosol ammonium-to-sulfate ratio and a lower aerosol pH (i.e. more acidic), based on Guo et al., 2018. In that paper, Guo et al. did a sensitivity study by applying observation-inferred Na+ in ISORROPIA in the southeast US, which improved the trend of ion ratios but did not change the trends in ISORROPIA-predicted pH using fixed Na+ concentration (Figure 6b in Guo et al., 2018).

The GEOS-Chem v12.1.1 does include NVCs (e.g. Na+, Ca2+, Mg2+ from anthropogenic and sea salt aerosol) in ISORROPIA (Pye et al., 2009, Pye et al., 2020), but they are not validated against measurements. In our study, the high H+ concentration (low aerosol pH) in August 2000-2008 is mainly caused by low NH3 emissions and high sulfate. The uncertainties associated with nonvolatile cations may lead to perturbations in aerosol pH and H+ concentration, which matters for the formation of IEPOX-SOA. However, as stated in Guo et al., the contribution of NVCs is relatively small (and may introduce more uncertainties), and aerosol pH is not very sensitive to NVCs, the extent to which the modeled OA would be impacted by considering NVCs might be small.

In the methods Section 2.2.1 at Line 144, we modify the sentence to: "GEOS-Chem v12.1.1 considers sulfate, nitrate, and ammonium from all sectors, and fine-mode Na+, Ca2+, Mg2+, Cl- from anthropogenic and sea salt sources, and employs the ISORROPIA II thermodynamic model (Fountoukis and Nenes, 2007; Pye et al., 2009; Song et al., 2018) to calculate aerosol water content and aerosol acidity (Pye et al., 2020)."

We mentioned the uncertainties associated with NVCs and acidity calculation in the results Section 3.3.2 at Line 373: "Due to the high uncertainty associated with the derived NH3 emission product and acidity calculation (Guo et al., 2015, 2018; Silvern et al., 2017; Song et al., 2018; Tao and Murphy, 2019), we conducted another simulation 'CT\_H01' that fix  $a_{H+}$  level at 0.1 mol/L when calculating IEPOX uptake rate ..."

In the discussion Section 4 at Line 496: "The NH3 emissions (which are critical for the calculation of aerosol acidity) are highly uncertain (Dammers et al., 2019), and the acidity calculation is further complicated by non-volatile cations (Guo et al., 2018) and meteorological conditions (Guo et al., 2015; Tao and Murphy, 2019)."

Minor comments.

4. Page 1, line 34 – The statement "Biogenic SOA may account for 60-100% of OA" is confusing, if you are going to call it biogenic SOA, but then discuss anthropogenic influence,

maybe call it something else, like biogenic-derived SOA or similar. Also, Kim et al., 2015 say that biogenic isoprene and monoterpenes account for 60% of OA, where do you get the 100% value?

Biogenic SOA is a term that usually refers to SOA formed from oxidation of biogenic volatile organic compounds. Over forest-covered regions biogenic SOA may account for up to 100% (e.g. Figure 2 in Xu et al., 2015).

To avoid confusion, we change the sentence to "Biogenic SOA (formed from atmospheric oxidation of BVOCs) may account for 60-100%..."

5. Page 5, line 70 – What about the role of NOx (and nitrate) in regulating aerosol water?

Nitrate contributes to a very small fraction of fine particles in the southeast US in summer. To be more comprehensive we add "NOx plays a complex role in regulating oxidation capacity, different oxidation pathways **and aerosol water content through aerosol nitrate.**"

6. Figure S1 – it is implied that the CSN sites are in Figure S1, which only shows SEARCH and IMPROVE.

Figure S1 has been revised. CSN sites are included as black dots. Due to the discontinuity of measurement protocol and techniques, 2000-2013 averaged OA concentrations are not shown for CSN sites.

7. Page 7, line 126 – Where does the baseline model agricultural NH3 come from?

In the US region, the baseline model NH3 emission (from all sectors including agriculture) come from the EPA's National Emission Inventory NEI11v1. In other regions NH3 emission come from the Community Emissions Data System (CEDS) inventory.

This information is included in Section 2.2.1 Line 124-128: "The global anthropogenic (including agricultural) emissions are from the Community Emissions Data System (CEDS) inventory, with the US region replaced by the EPA's National Emission Inventory for 2011 (NEI11v1). The monthly mean anthropogenic emissions of CO, SO2, NOx, NH3, VOCs, OC and black carbon are scaled to the year 2011 using the ratio of EPA's national annual emission totals from 2000 to 2013 (Travis et al., 2016)."

8. Page 7 line 126 - Do you mean you scale the emissions from 2011 backwards to 2000 and forwards to 2013 using the annual emission totals from EPA? What do you mean by "mapped over 0.1x0.1"

Yes. The other years' emissions are scaled using ratios of EPA's annual emission total in each year relative to the emission in 2011. The emissions have a horizontal resolution of  $0.1^{\circ}x0.1^{\circ}$ . We removed the words "mapped over 0.1x0.1" to avoid confusion as this information is not necessary.

We change the sentence to: "The monthly mean anthropogenic emissions of CO, SO2, NOx, NH3, VOCs, OC and black carbon are scaled to the year 2011 using the ratio of EPA's national annual emission totals from 2000 to 2013 (Travis et al., 2016)."

9. Page 8, line 134 – The GEOS-Chem complex scheme is also described in great detail by Pai et al., 2020. "An evaluation of global organic aerosol schemes using airborne observations"

The Pai et al. 2020 has been added here as a reference.

10. Page 8, line 148. It is confusing to dive in and describe sensitivity studies without showing us the problem in Figure 1. I don't think it would be out of place to say that the default model, shown in Figure 1, overestimates the trend in organic aerosol largely due to ... and we run 4 sensitivity studies to address this discrepancy...

We add: "The default modeled OA shows a stronger decreasing trend from 2000 to 2013, and a large month-to-month variability in early 2000s, different from the observations (more details in Figure 1, 2 and Section 3.1). To address this model-observation discrepancy, we do four sets of  $2^{\circ} \times 2.5^{\circ}$  simulations: ..."

11. Page 9, line 154 – It would be helpful to tell the reader that these sensitivity simulations will be described in Sections 2.2.2 etc..

We add: "The sensitivity simulations are further explained in Section 2.2.2-2.2.3 and Section 3.3."

12. Page 9, line 163 – do you mean at "the" or "from the" SOAS2013 campaign (citation?)

We change the sentence to: "The default IEPOX-SOA mechanism in GEOS-Chem uses aerosolphase reaction rates from laboratory chamber studies with pure acidic inorganic particles (Gaston et al., 2014; Riedel et al., 2015), and a representative effective Henry's law constant obtained by matching the model to the observations **from the** SOAS2013 campaign (**Marais et al., 2016**), to estimate the reactive uptake coefficient  $\gamma_{IEPOX}$ ."

13. Do the authors expect that not including non-volatile cations in ISOROPPIA could impact their findings?

See response to comment #3.

14. Page 10, line 178 – Can you provide some evidence/citations for us that there might be such a coating effect? Or is the ethylene glycol supposed to be representative enough of real organic aerosol coating ammonium bisulfate? Please just clarify a little more here.

Here we add a sentence: "In the real atmosphere, inorganic aerosol is generally internally mixed with other organics. The presence of an organic coating may alter the aerosol properties and suppress the uptake of IEPOX onto acidified sulfate aerosol (Anttila et al., 2006; Gaston et al., 2014)."

The reasoning is of adding a coating effect is also explained at the beginning of Section 3.3.1 (L330-337): "The modeled IEPOX-SOA shows a much stronger sensitivity to aerosol acidity than suggested by field observations, which found weak or no correlation between observed IEPOX-SOA and derived aerosol acidity (Budisulistiorini et al., 2015; Lin et al., 2013; Worton et al., 2013; Xu et al., 2015b). Lack of consideration of organic coating effect may provide one possible explanation. In the real atmosphere, inorganic aerosol is generally internally mixed with other organics (Anttila et al., 2006; Murphy et al., 2006). The presence of an organic coating may alter the solubility and diffusion properties at the surface of inorganic particles and diminish further uptake of IEPOX."

We also add at Line 180 about the limitation of ethylene glycol and the uncertainties with the linear fitted function: "In the real atmosphere when inorganic cores are coated with more viscous SOA (Zhang et al., 2018b), coating effect may be stronger because ethylene glycol is a low viscosity material. However, this simplified linear function does not consider the decreased viscosity and reduced coating effect at higher RH conditions (which is common in summertime southeast US) (Gaston et al., 2014; Zhang et al., 2018b), and prevents further IEPOX uptake when the mass fraction of OA ( $\chi_{org}$ ) is larger than 0.7, therefore this linear function may mimic a strong coating effect even though ethylene glycol is less viscous than real atmospheric SOA. The uncertainties need to be addressed in further studies with a more realistic coating parameterization (Li et al., 2020; Schmedding et al., 2019; Zhang et al., 2019b)."

15. Page 10, line 189 - Please elaborate a little more on the findings of Jo et al., 2019 since this seems important.

Jo et al. is also a modeling study. Here this effect is because the standard GEOS-Chem assumes no organic coating, and the aerosol surface and radius only consider inorganics (sulfateammonium-nitrate). By considering organic coating, the total aerosol surface and radius both increase which tends to increase the IEPOX uptake rate constant. Jo et al. found in some conditions (high aerosol pH and high IEPOX diffusion coefficient) the IEPOX uptake rate may even increase. The coating parameterizations in Jo et al. and in our study are different.

We remove the statement "consistent with another study (Jo et al., 2019)" to avoid confusion. We modify the sentence as: "The default GEOS-Chem with no organic coating calculates surface area of inorganic aerosol (Jo et al., 2019). By adding the coating effect, the increased particle radius  $R_p$  and surface area  $S_a$  of the mixed particle will partially offset (but does not outweigh) the impact of reduced reaction probability  $\gamma_{IEPOX modified}$ ."

16. In Section 2.2.3 – How do your emissions results compare to the NH3 emissions derived from wet deposition in Paulot et al., 2014? Paulot F., Jacob, D.J., Pinder R.W., Bash J.O., Travis, K., Henze D.K., Ammonia emissions in the United States, Europe, and China derived by high-resolution inversion of ammonium wet deposition data: Interpretation with a new agricultural emissions inventory (MASAGE\_NH3), J. Geophys. Res., 119, 4,343-4,364, 2014.

This CrIS-derived product has higher NH3 emissions than the estimates from Paulot et al. (2014) (see Table 1 and more details in Cao et al., 2020).

17. Page 13, line 241 – Do the sulfate trends refer to Figure 2? If so, please reference it here.

We change to: "Compared to the slow decrease in OA, a faster declining trend is found for sulfate from IMPROVE (-6.9%/year) and SEARCH (-6.7%/year) for the same period (**Figure 2**)."

18. Page 11, line 199 – Is this due to improved farming practices as output increases? Or some other explanation? I am surprised NH3 is not increasing and would assume other readers might appreciate a little more detail.

NH3 emissions have no significant trend in 2000-2013. The NH3 gas concentration is increasing based on other observations, but it is not what we discussed here.

To avoid confusion, we add "There is no significant trend **of NH3 emissions** from 2000 to 2013 (Figure S4), consistent with other studies suggesting nearly constant NH3 emissions from 2001 to 2014 (Butler et al., 2016)."

19. Section2.2.3 – It would help the reader if there was a plot of the bottom-up and top-down NH3 emissions.

We add a Figure S3 to show the map of the default NEI11 (bottom-up) and the CrIS-derived (top-down) NH3 emissions in June, July and August (averaged over 2000-2013).

20. Page11, line 208 – What surface observations? Please clarify what you mean, are there separate independent NH3 surface observations you compare against? Again, a plot would be useful here. Same for the seasonal cycle. These can be in the supplement according to the author's preference.

The top-down NH3 emissions are derived and validated in another paper Cao et al., 2020. Here we use the new NH3 emissions as a sensitivity test to show how sensitive IEPOX-SOA is to aerosol acidity and NH3 emissions. We do not include any validation plots because they might deviate from the main focus of this paper. Instead we add the citation and add the text:

"The CrIS-derived NH3 emissions have been validated against surface observations of NH3 concentration from the Ammonia Monitoring Network (AMoN) and NH4+ wet deposition measurements from the National Atmospheric Deposition Program (NADP). More details can be found in Cao et al. (2020)."

21. Page11, line217 – For clarity, please specific that these are "model" IEPOX-SOA, sulfate aerosol etc.

We change to: "In this study we did a multivariate regression analysis of **modeled** monthly IEPOX-SOA ( $\mu$ g/m3) against **modeled** sulfate aerosol ( $\mu$ g/m3), aerosol acidity  $a_{H^+}$  (mol/L) and isoprene emission (*ISOP*emis mg/m2/hr)."

22. Page 13, line 244 – Why not sample the model at the locations of the sites? Would that make a difference to your trend?

The modeled summertime OA, if sampled at the locations of sites, has an average of 6.9  $\mu$ g/m3 and a trend of 5.0%/year. These results are similar to the model results averaged over the whole southeast US domain (an average of 6.7  $\mu$ g/m3 and a trend of 4.9%/year). For simplicity and to be consistent with other analysis in this study, we only show the model results averaged over the whole domain here.

Here we add the text: "Modeling results are averaged over the domain  $[29^{\circ} \sim 37^{\circ}N, 74^{\circ} \sim 96^{\circ}W]$  excluding ocean grid cells (Figure S1). The 2000-2013 JJA-averaged OA from the default model is 6.7 µg/m3, higher than OA from IMPROVE and SEARCH. Modeled total OA mass decreases at a rate of 4.9%/year, about 1.9 (1.6) times faster than IMPROVE (SEARCH) OA (student's t-test p<0.001). By sampling the model results at the locations of the IMPROVE and SEARCH sites, the modeled summertime OA has an average of 6.9 µg/m3 and a trend of 5.0%/year, similar to the model results averaged over the whole southeast US domain. For simplicity, we show only the domain-averaged model results in all figures and analysis."

23. Figure 1 – can you explain the periods of enhanced OA around 2006 and 2011?

The enhanced OA in 2006 and 2011 is due to high isoprene emissions (Figure 3). In Section 3.2 Line 326 we change to: "Isoprene emission contributes to the overall interannual variability, for example leading to the relatively low IEPOX-SOA in 2003-2004 and the peaks in 2000, 2006 and 2011 (Figure 3)."

24. Figure 2 – It could be helpful to include the trends on this figure similar to Figure 1.

Figure 2 has been modified to include trends of OA and sulfate.

25. Page 15, line 302 – Figure 3 is difficult to interpret, please provide a little more description of what is shown here, particularly what is mean by "divided by 1 standard deviations."

We add the following descriptions: "Figure 3 shows the standardized monthly surface IEPOX-SOA concentration, sulfate concentration, aerosol H+ activity and isoprene emission from the default model. For each variable, the monthly gridded data has been first averaged over the southeast US. Then, we calculate the one standard deviation of all monthly data (June, July and August data from 2000 to 2013). Finally, the domain-averaged monthly data has been divided by its standard deviation, so the variables are standardized to be unitless and their variability can be compared directly."

26. Page 16, line 325 – Is isoprene interannual variability explained by temperature? Temperature is an important driver of aerosol partitioning, should it not be included in Table S1 and Figure 3? Does temperature explain any of the interannual variation in Figure 1?

Yes, temperature is an important factor controlling the variability of isoprene emissions and therefore SOA. In GEOS-Chem, SOA formed by aqueous-phase uptake of isoprene products

(which dominates the total OA in the southeast US as shown in Figure 1) is assumed to be nonvolatile, and temperature plays a minor role in these processes. The monoterpene-derived SOA is considered to be semi-volatile and formed by reversible gas-aerosol partitioning, which is influenced by temperature. But monoterpene-SOA contributes to a smaller fraction of OA (Figure 1). Here we focus on the variability of IEPOX-SOA (which is non-volatile), and we already consider isoprene emission as a main driver. Therefore, we do not consider temperature as an additional independent driver of IEPOX-SOA in Table S1 and Figure 3.

Earlier at Line 295 we add: "We use the GEOS-Chem model here to examine the simulated IEPOX-SOA dependence on sulfate, aerosol acidity, and emissions of isoprene which produce IEPOX at high yields under low-NOx conditions (Paulot et al., 2009). **Temperature impacts the formation of IEPOX-SOA mainly through regulating isoprene emissions but does not influence partitioning as IEPOX-SOA is treated as non-volatile in GEOS-Chem. Therefore, temperature is not examined as another driver in addition to isoprene emissions."**

27. Page 18, line 376 – I assume that the H+ value during 2013SOAS was calculated with a thermodynamic model, although constrained by observations. Please clarify.

Yes. We change the wording here to: "we conducted another simulation 'CT\_H01' that fix  $a_{H^+}$  level at 0.1 mol/L when calculating IEPOX uptake rate, corresponding to the **predicted**  $a_{H^+}$  value (**constrained by observations**) during the 2013 SOAS campaign (Weber et al., 2016)."

28. Page 19, line 378 – Can you comment on why the model agrees much better with SOAS2013 observations and 2013 OA in Figure 1, and what the limitations are of testing your revised IEPOX uptake at the low end of the SO2 emissions trend?

Figure 1 is not about SOAS2013 campaign data. Figure 1 shows the comparison between the default model results and the long-term filter measurements of OA from surface networks IMPROVE and SEARCH. Our revised IEPOX uptake agrees well with the long-term surface measurement in 2000-2013 (Figure 4C and 4D), which is the main focus of our paper. The 2000-2013 period covers a broad range of SO2 emissions. The revised IEPOX-SOA also compares well with the SOAS2013 data (Figure S6).

In Line 378, we add: "The two simulations, CT\_newNH3 and CT\_H01, yield similar long-term trends of IEPOX-SOA in the southeast US (Figure S5), **and they agree better with the long-term surface OA measurements from IMPROVE and SEARCH than the default model (Figure 4C and 4D).** For the SOAS2013 campaign, the CT\_H01 scheme simulates an average IEPOX-SOA concentration of 0.74  $\mu$ g/m3, similar to 0.81  $\mu$ g/m3 in the default model, and agrees well with the two independent Aerosol Mass Spectrometer measurements (0.97  $\mu$ g/m3 from obs\_GT and 0.68  $\mu$ g/m3 from obs\_CU, see daily time series in Figure S6)."

29. General comment – why not consider trends in ammonium? Particularly to help support your improved NH3 emissions scenario?

The long-term observation network IMPROVE does not have measurements of ammonium aerosol. SEARCH shows a downward trend of ammonium aerosol, similar to the trend of sulfate

(Silvern et al., 2017). In the model ammonium (aerosol  $NH_4^+$ ) is highly correlated with sulfate and has a very similar trend as sulfate, so we do not show the trends in ammonium here. The month-to-month variability of  $NH_3$  emissions is what matters that leads to high aerosol  $H^+$  in August in early 2000s (and therefore IEPOX-SOA). The  $NH_3$  emissions do not have significant long-term trends based on EPA's annual emission totals.

We add the trends of aerosol NH4+/SO42- ratio to Figure S4, and change the text in Section 3.2 around L308: "The default NH3 emissions from NEI11v1 suggest no significant long-term trend from 2000 to 2013. In general, ammonium aerosol is strongly correlated with sulfate and has a similar declining trend as sulfate (Silvern et al., 2017). However, the NH3 emissions in August are about 25% lower than in June and July (Figure S4). As a result, in August before 2008, the aerosol NH4+/SO42- ratio is smaller (Figure S4) and the  $a_{H^+}$  is up to 3 times higher than June, leading to high production of IEPOX-SOA in August."

30. Page 22, line 467 – By gas-aerosol partitioning, do you mean reversible uptake of isoprene oxidation products? Could you be more specific on this?

a. Generally, the final paragraph that goes through potential reasons for the remaining model discrepancies is very useful but needs additional explanation. The sentence starting "Despite a large MMV in IEPOX-SOA is confusing, please rephrase to more clearly state your meaning.

For the above questions, we change the text as follows: "Such discrepancies may suggest a more important role of SOA pathways that are less dependent on inorganic aerosol, such as **terpene-SOA formed by reversible** gas-aerosol partitioning. **Terpene-SOA is included in GEOS-Chem (yellow color in Figure 1), and contributes to 8~24% of total OA, which might be underestimated compared to recent field studies. Xu et al. (2015a) finds a large MMV in IEPOX-SOA, but the less-oxidized oxygenated OA (LO-OOA, an indicator for freshly-formed monoterpene SOA) and the more-oxidized oxygenated OA (MO-OOA, also likely from biogenic sources) have little MMV in summer months, and** they contribute to more than 50% of total OA mass in the southeast US (Xu et al., 2018). The important role of monoterpenes SOA is also confirmed by molecular level characterization of organic aerosols (Zhang et al., 2018a)."

b. Is monoterpene SOA included in the GEOS-Chem simulations? If so, is the lack of MMV captured by the model? I am generally just confused by the discussion here. Are you trying to say that other SOA pathways, not in the model, might have less MMV and therefore improve model performance of the long-term trend?

Yes, monoterpene SOA is included in GEOS-Chem (see changes above). In Figure 1 yellow color represents terpene SOA (mostly monoterpene and also some sesquiterpene, see Section 2.2.1 Line 137). Terpene SOA and anthropogenic SOA are referred to as dry SOA (Line 53). They have small MMV (Figure 2B) but they only contribute to a small fraction of total OA especially when comparing to IEPOX-SOA in early 2000s (Figure 1). We would like to strengthen that monoterpene SOA may be underestimated in GEOS-Chem, compared to the field measurements (Xu et al., 2018; Zhang et al., 2018), see Line 508-515. Other pathways that are less dependent on sulfate and aerosol acidity (and probably less MMV), not in the model now, may improve the model performance and need further examination (Line 515-525).

**Reference:**

Anttila, T., Kiendler-Scharr, A., Tillmann, R. and Mentel, T. F.: On the reactive uptake of gaseous compounds by organic-coated aqueous aerosols: Theoretical analysis and application to the heterogeneous hydrolysis of N2O5, J. Phys. Chem. A, 110(35), 10435–10443, doi:10.1021/jp062403c, 2006.

Cao, H., Henze, D. K., Shephard, M. W., Dammers, E., Cady-Pereira, K., Alvarado, M., Lonsdale, C., Luo, G., Yu, F., Zhu, L., Danielson, C. G. and Edgerton E. S., accepted by Environ. Res. Lett., 2020.

[revised manuscript text omitted]

Zhang, Y., Chen, Y., Lambe, A. T., Olson, N. E., Lei, Z., Craig, R. L., Zhang, Z., Gold, A., Onasch, T. B., Jayne, J. T., Worsnop, D. R., Gaston, C. J., Thornton, J. A., Vizuete, W., Ault, A. P. and Surratt, J. D.: Effect of the Aerosol-Phase State on Secondary Organic Aerosol Formation from the Reactive Uptake of Isoprene-Derived Epoxydiols (IEPOX), Environ. Sci. Technol. Lett., 5(3), 167–174, doi:10.1021/acs.estlett.8b00044, 2018b.

**Response to Reviewer #2**

We are grateful to the reviewer for the helpful comments and guidance that have led to important improvements of the original manuscript. Our point-by-point responses are listed below. Reviewer's comments are in black font, and authors' responses are in dark blue. Page and line numbers refer to the discussion paper acp-2020-575. The revised figures and Supplementary Information are attached in the end.

Summary: The author compared the long-term trend of organic aerosol and sulfate mass loading from field measurement with GEOS-Chem simulations in this manuscript, with the model showing a steeper decrease in the OA mass loading and larger monthto-month variability than the field data. The long-term trend of sulfate, on the other hand, was well captured by the model when comparing with the field data. By applying coating effect, constant aerosol acidity, and a different NH3 emission product, the modeling results match the field data better, suggesting further study is needed to address the weak dependence of OA formation and sulfate. The manuscript is overall clearly written, but may need to address the following aspects before publishing.

First, the author did a good job explaining the modeling results in Section 3, but it was not very clear at certain sections which modeling results agree with the field measurement and which do not. The author may need to improve clarity about the modeling-field data comparisons when describing the modeling results, especially in line 251 when the author discusses that the contribution of IEPOX-SOA to total OA mass decreases from 61% to 28% from the early 2000s to 2013. Does this modeling result agree with previous field measurements (such as the results from Xu. et al. 2015 and Budisulistiorini, et al. 2016)? Maybe the author can include a sentence or two to compare the modeling data with the field data.

Section 3.1 mainly compares modeling results with long-term surface filter measurements from IMPRAOVE and SEARCH networks, focusing on the long-term trend and month-to-month variability of OA (Figure 1 and 2). We also compare the modeling results with the SOAS2013 field campaign data later in Section 3.3.2 Line 379-385 and Figure S6.

Here we add a sentence about the contribution of IEPOX-SOA: "The contribution of IEPOX-SOA to total OA mass decreases from 61% in the early 2000s to 28% in 2013. The simulated IEPOX-SOA in 2013 compares well with previous field studies which suggested that IEPOX-SOA contributed to 18~40% in southeast US sites in summer 2013 (Budisulistiorini et al., 2016; Xu et al., 2015)."

The manuscript also concluded that coating can improve the modeling result because thinner coating may enhance the formation of IEPOX-SOA. There are a few papers that also measured/modelled the effects of SOA coating on the formation of IEPOX-SOA. For instance, Zhang et al. 2018, Jo et al., 2019, and the subsequent study by Schmedding et al. 2019 discussed the effects of pre-existing coating on the formation of IEPOX-SOA. Does the result in this manuscript using ethylene glycol agree with previous studies using authentic SOA? In addition, Jo et al. 2019 shows that the uptake of IEPOX would increase with increasing coating for most of the situations using GEOS-Chem due to increasing surface tension, contrary to a decreasing

uptake with coating effect in this manuscript. Could the author explain why the trends are different in these two studies?

Compared to the default GEOS-Chem with no coating effect at all, adding a coating effect reduces (not enhances) the production of IEPOX-SOA in the summertime southeast US (Figure 4C, orange line as compared to magenta line). Jo et al. applied the parameterizations from Zhang et al., 2018 (using monoterpene-SOA as a coating material) and is different from what we used here. In Gaston et al. (2014) which we applied in this study, ethylene glycol is a relatively low viscosity material, but the simplified linear function fitted using RH=50% conditions may mimic a strong coating effect, because it does not consider the reduced viscosity and weaker coating effect at higher RH conditions, and it assumes no IEPOX uptake when the fraction of organics is higher than 70%. As a result, the added aerosol surface and particle radius does not overweigh the impact of reduced uptake coefficient when considering coating.

In Section 2.2.2 Line 180 we add: "In the real atmosphere when inorganic cores are coated with more viscous SOA (Zhang et al., 2018b), coating effect may be stronger because ethylene glycol is a low viscosity material. However, this simplified linear function does not consider the decreased viscosity and reduced coating effect at higher RH conditions (which is common in summertime southeast US) (Gaston et al., 2014; Zhang et al., 2018b), and prevents further IEPOX uptake when the mass fraction of OA ( $\chi_{org}$ ) is larger than 0.7, therefore this linear function may mimic a strong coating effect even though ethylene glycol is less viscous than real atmospheric SOA. The uncertainties need to be addressed in further studies with a more realistic coating parameterization (Li et al., 2020; Schmedding et al., 2019; Zhang et al., 2019b). We assume all OA is coated outside the inorganic aerosol core when calculating the IEPOX reactive uptake. The default GEOS-Chem with no organic coating calculates surface area of inorganic aerosol. By adding the coating effect, the increased particle radius  $R_p$  and surface area  $S_a$  of the mixed particle will partially offset (but does not overweigh) the impact of reduced reaction probability  $\gamma_{IEPOX\_modified}$ ."

The manuscript used a fixed acidity to reduce the modelled month-to-month variation of IEPOX-SOA so that the results match better with the field data. It seems this large month-to-month variation in modeling only appeared before 2008, while the month-to-month variation decreased significantly after 2008 even with the default modeling scenario. Can the author explain why there is such a large difference in the month-to-month variation before and after 2008 in the default modeling scenario? Was it due to difference in NH3 emission inventory before and after 2008 or other reasons?

We explained the large MMV before 2008 in Section 3.2. The main idea is that when sulfate is high before 2008, the relatively low  $NH_3$  emission in August is insufficient relative to sulfate and leads to a much higher aerosol H+ (than in June and July), which then leads to high IEPOX-SOA. After 2008 when sulfate becomes small, aerosol H+ becomes less sensitive to the supply of  $NH_3$ , so even there are still monthly differences in  $NH_3$  emissions, the MMV of IEPOX-SOA is smaller.

In Section 2.2.2 At Line 301-303 we have the explanation: "Prior to 2008, IEPOX-SOA production is largely enhanced by abundant sulfate. Due to this high level of sulfate (about >4

 $\mu$ g/m3), the modeled aerosol acidity becomes particularly sensitive to variations in NH3 emissions. ... The NH3 emissions in August are about 25% lower than in June and July (Figure S4). As a result, in August before 2008, the aerosol NH4+/SO42- ratio is smaller (Figure S4) and aH+ is up to 3 times higher than June (Figure 4B), leading to high production of IEPOX-SOA in August. Both sulfate and aerosol acidity appear to be the dominant contributors to MMV of OA during this period. After 2008, IEPOX-SOA formation is substantially suppressed, due to small SO2 emissions and low modeled aerosol acidity aH+ with small monthly variability."

In addition, after updating the NH3 inventory with the new emission inventory from CrIS satellite observation, the author stated in line 384-385 that this scenario (CT\_newNH3) performed a bit worse than the fixed acidity scenario (CT\_H01). I wonder whether that was due to coating effects not correctly represented by using ethylene glycol rather than the results from authentic coatings. Besides the paper mentioned above, Li et al. 2020 and Zhang et al. also used simplified equations that can estimate the phase state of a few IEPOX-SOA species that might be helpful in performing future modeling.

Thank you for the suggestions. The CT\_newNH3 simulation may be influenced by both the simplified coating scheme as well as uncertainties associated with NH3 emissions. In the text we have:

"The CT\_newNH3 scheme simulates an average IEPOX-SOA concentration of 0.34  $\mu$ g/m3, lower than the observation and the other models by a factor of >2, due to both the simplified coating effect and small aerosol  $a_{H^+}$  values ( $a_{H^+}$ <0.1mol/L, Figure 4B)." The Li et al. and Zhang et al. papers have been added to the model description Section 2.2.2 (see response to the second comment).

Minor comments: Why would there be a large increase of the default IEPOX-SOA during 2005-2008? The manuscript mentioned about higher correlation of IEPOX-SOA with acidity during this period of time. Was this abrupt increase of IEPOX-SOA caused by lower NH3 emissions between 2005-2008?

The high IEPOX-SOA in 2005-2007 is a result of both high sulfate, high aerosol acidity in August and high isoprene emissions.

In Section 3.2 at Line 314, we add: "The high IEPOX-SOA in 2000-2001 and 2005-2007 is a result of high sulfate aerosol, high aerosol acidity due to low NH3 supply relative to high sulfate, and high isoprene emissions during these periods (Figure 3, Figure 4B)."

Line 171: There are multiple papers discussing about different Henry's law constants for IEPOX. The author did discuss in line 439 but probably should include other relevant papers, such as Woo et al., 2015, Budisulistiorini et al., 2016, Pye et al., 2017, and Zhang et al., 2018.

Thank you for the suggestions. We add at Line 439:

"the Henry's law constant for IEPOX uptake was tuned using measurements from the SOAS2013 campaign and was found to be  $1.7 \times 10^7$  M/atm, 10 times smaller than suggested by

Gaston et al. (2014) based on laboratory experiments and about half of the suggested value (3×107 M/atm) in some other studies (Budisulistiorini et al., 2017; Nguyen et al., 2014; Pye et al., 2017; Woo and McNeill, 2015; Zhang et al., 2018b)."

Line 477-478: One other potential mechanism I can think of is the non-linear feedback between sulfate and IEPOX-SOA production discussed in recent studies. For instance, Riva et al. 2019 and Zhang et al. both show that IEPOX-SOA fraction could and sulfate are nonlinear due to chemical reactions, acidity, and the coating effects of IEPOX-SOA are intertwined and nonlinear due to the formation of organosulfates.

Thank you for the suggestions. We mentioned the non-linearity at Line 408-410, and the Zhang et al. paper has been added as a reference now:

"Recent studies (Riva et al., 2019) suggested that the IEPOX-SOA production per unit mass of sulfate likely increases with decreasing sulfate due to changes in aerosol properties, such as acidity, morphology, phase state and viscosity, as well as formation of organosulfates, suggesting non-linearity between IEPOX-SOA and sulfate (Riva et al., 2019; Zhang et al., 2019)."

**Long-term observational constraints of organic aerosol dependence on inorganic species in the southeast US**

Yiqi Zheng1,2,\*, Joel A. Thornton3, Nga Lee Ng4,5,6, Hansen Cao7, Daven K. Henze7, Erin E. McDuffie8,9, Weiwei Hu10,11, Jose L. Jimenez11, Eloise A. Marais12,13, Eric Edgerton14, Jingqiu Mao1,2,\*

[revised manuscript text omitted]

**168 **2.2.2 Coating**

The default IEPOX-SOA mechanism in GEOS-Chem uses aerosol-phase reaction rates from
laboratory chamber studies with pure acidic inorganic particles (Gaston et al., 2014; Riedel et al.,
2015), and a representative effective Henry's law constant obtained by matching the model to the
observations from the SOAS2013 campaign\_(Marais et al., 2016), to estimate the reactive uptake
coefficient *piEPOX*. In the default scheme, *piEPOX* is calculated as follows:

174
$$\frac{1}{\gamma_{IEPOX}} = \frac{R_p \omega}{4D_g} + \frac{1}{\alpha} + \frac{1}{\Gamma_{aq}}$$

175
$$\Gamma_{aq} = \frac{4VRTH_{aq}k_{aq}}{S_a\omega}$$

176 Where  $R_p$  is the particle radius of the inorganic sulfate-nitrate-ammonium particle (cm),  $\omega$  is the 177 mean molecular speed (cm/s),  $D_g$  is the gas-phase diffusion coefficient (0.1 cm2/s),  $\alpha$  is the mass 178 accommodation coefficient ( $\alpha$ =0.1),  $S_a$  is the total (wet) particle surface area (cm2/cm3), V is the 179 total (wet) particle volume (cm3/cm3), R is the ideal gas constant (L atm/mol/K), T is temperature 180 (K),  $H_{aq}$  is the Henry's law coefficient (1.7×107 M/atm), and  $k_{aq}$  is the first-order reaction rate 181 constant  $(s^{-1})$ :  $k_{aq} = k_{H^+}[H^+] + k_{nuc}[nuc]a_{H^+} + k_{ga}[ga]$ 182 where  $k_{H^+}$  (=0.036 M-1s-1),  $k_{nuc}$  (=2×10-4 M-1s-1) and  $k_{ga}$  (=7.3×10-4 M-1s-1) are the reaction 183 184 rates due to acid-catalyzed ring-opening, presence of nucleophiles (including nitrate and sulfate) 185 and presence of bisulfate acids, respectively (Gaston et al., 2014; Marais et al., 2016). 186 187 In the real atmosphere, inorganic aerosol is generally internally mixed with other organics. The 188 presence of an organic coating may alter the aerosol properties and suppress the uptake of 189 IEPOX onto acidified sulfate aerosol (Anttila et al., 2006; Gaston et al., 2014). We implement a 190 linear coating effect for the IEPOX-SOA formation. The coating effect is fitted using laboratory-191 derived values of *y*IEPOX on particles containing both ammonium bisulfate and ethylene glycol 192 under RH=50% conditions (Gaston et al., 2014). In the coating scheme,  $\gamma'_{IEPOX}$  is calculated as 193 above with  $R_p$ , V and  $S_a$  updated considering OA coated outside the inorganic core. Then, the 194 fitted function is applied to modify  $\gamma'_{IEPOX}$ : 195  $\gamma_{IEPOX\_modified} = \gamma'_{IEPOX} \times (1 - 1.3 \times \chi_{org})$ 196 where  $\chi_{org}$  is the mass fraction of OA in the mixed particle including both the inorganic aerosol 197 and OA. When  $\chi_{org} > 0.7$ , the IEPOX uptake will be terminated, i.e.  $\gamma_{IEPOX\_modified} = 0$ . In the 198 real atmosphere when inorganic cores are coated with more viscous SOA (Zhang et al., 2018b), 199 coating effect may be stronger because ethylene glycol is a low viscosity material. However, this simplified linear function does not consider the decreased viscosity and reduced coating effect at 200 201 higher RH conditions (which is common in summertime southeast US) (Gaston et al., 2014; 202 Zhang et al., 2018b), and prevents further IEPOX uptake when the mass fraction of OA ( $\chi_{org}$ ) is

| 203 | larger than 0.7, therefore this linear function may mimic a strong coating effect even though                                 |
|-----|-------------------------------------------------------------------------------------------------------------------------------|
| 204 | ethylene glycol is less viscous than real atmospheric SOA. The uncertainties need to be                                       |
| 205 | addressed in further studies with a more realistic coating parameterization (Li et al., 2020;                                 |
| 206 | Schmedding et al., 2019; Zhang et al., 2019b). We assume all OA is coated outside the inorganic                               |
| 207 | aerosol core when calculating the IEPOX reactive uptake. The default GEOS-Chem with no                                        |
| 208 | organic coating calculates surface area of inorganic aerosol (Jo et al., 2019). By adding the                                 |
| 209 | coating effect , the increased particle radius $R_p$ and surface area $S_a$ of the mixed particle will                 |
| 210 | partially offset (but does not outweigh) the impact of reduced reaction probability                                           |
| 211 | YIEPOX_modified:                                                                                                              |
| 212 |                                                                                                                               |
| 213 | 2.2.3 Satellite-derived NH 3 emissions                                                                             |
| 214 | We use the Cross-track Infrared Sounder (CrIS) satellite-derived NH3 emissions_(Cao et al.,                                   |
| 215 | 2020) in a sensitivity test in this study. The top-down monthly NH 3 emissions over the                            |
| 216 | contiguous US at $0.25^{\circ} \times 0.3125^{\circ}$ latitude by longitude are derived from CrIS v1.5                        |
| 217 | measurements of $NH_3$ profiles (Shephard and Cady-Pereira, 2015) for the year 2014 through a                                 |
| 218 | 4D-Var approach using GEOS-Chem and its adjoint model (Henze et al., 2007). The CrIS-                                         |
| 219 | derived emissions are then regridded to $0.1^{\circ} \times 0.1^{\circ}$ to replace the default NEI11 emissions for the       |
| 220 | year 2011 and applied the same annual scaling factors for 2000-2013. The default NEI11 and                                    |
| 221 | CrIS-derived NH 3 emissions averaged over 2000-2013 are compared in Figure S3. There is no                         |
| 222 | significant trend of <math>NH_3</math> emissions from 2000 to 2013 (Figure S 4 ), consistent with other studies |
| 223 | suggesting nearly constant NH 3 emissions from 2001 to 2014 (Butler et al., 2016). The CrIS-                       |
| 224 | derived emissions used the HTAPv2 emissions inventory as the prior emissions, which is based                                  |
| 225 | on the 2008 NEI emissions over the US (Janssens-Maenhout et al., 2015). The CrIS-derived NH 3                      |
| 1   |                                                                                                                               |

| 226                                                                                                                                          | emissions have been validated against surface observations of $NH_3$ concentration from the                                                                                                                                                                                                                                                                                                                                                                                                                                                                                                                                                                                                                                                                                                                                    |                                                                                              |
|----------------------------------------------------------------------------------------------------------------------------------------------|--------------------------------------------------------------------------------------------------------------------------------------------------------------------------------------------------------------------------------------------------------------------------------------------------------------------------------------------------------------------------------------------------------------------------------------------------------------------------------------------------------------------------------------------------------------------------------------------------------------------------------------------------------------------------------------------------------------------------------------------------------------------------------------------------------------------------------|----------------------------------------------------------------------------------------------|
| 227                                                                                                                                          | Ammonia Monitoring Network (AMoN) and wet deposition measurements from the National                                                                                                                                                                                                                                                                                                                                                                                                                                                                                                                                                                                                                                                                                                                                            |                                                                                              |
| 228                                                                                                                                          | Atmospheric Deposition Program (NADP). More details can be found in Cao et al. (2020). Using                                                                                                                                                                                                                                                                                                                                                                                                                                                                                                                                                                                                                                                                                                                                   | Deleted: The top-down annual mean emiss                                               |
| 229                                                                                                                                          | the top-down emissions in GEOS-Chem increases the correlation coefficient $(r)$ between                                                                                                                                                                                                                                                                                                                                                                                                                                                                                                                                                                                                                                                                                                                                        | emissions underestimate agricultural emiss
springtime fertilizer and livestock sources of |
| 230                                                                                                                                          | modeled monthly mean $NH_3$ concentration and surface observations from 0.74 to 0.93 and                                                                                                                                                                                                                                                                                                                                                                                                                                                                                                                                                                                                                                                                                                                                       | emissions were found in the Central Valley
Minnesota, northern Iowa and southeast No      |
| 231                                                                                                                                          | reduces the normalized mean bias of domain-averaged annual mean simulated NH3 by a factor of                                                                                                                                                                                                                                                                                                                                                                                                                                                                                                                                                                                                                                                                                                                                   | during warm months.                                                                          |
| 232                                                                                                                                          | 1.9. The seasonal cycle of simulated wet $NH_4^+$ deposition is also improved ( r increased from                                                                                                                                                                                                                                                                                                                                                                                                                                                                                                                                                                                                                                                                                                                        |                                                                                              |
| 233                                                                                                                                          | 0.70 to 0.86), but the normalized mean bias of domain-averaged annual simulated wet $\mathrm{NH_4^+}$                                                                                                                                                                                                                                                                                                                                                                                                                                                                                                                                                                                                                                                                                                                          |                                                                                              |
| 234                                                                                                                                          | increases from 0.34 to 0.96 due to overly strong wet scavenging in the model. The latter issue                                                                                                                                                                                                                                                                                                                                                                                                                                                                                                                                                                                                                                                                                                                                 |                                                                                              |
| 235                                                                                                                                          | was ultimately resolved in Cao et al. (2020) and the final top-down emissions reported therein                                                                                                                                                                                                                                                                                                                                                                                                                                                                                                                                                                                                                                                                                                                                 |                                                                                              |
| 236                                                                                                                                          | differ from those reported here; nevertheless, the emissions estimates used here provide a                                                                                                                                                                                                                                                                                                                                                                                                                                                                                                                                                                                                                                                                                                                                     |                                                                                              |
|                                                                                                                                              |                                                                                                                                                                                                                                                                                                                                                                                                                                                                                                                                                                                                                                                                                                                                                                                                                                |                                                                                              |
| 237                                                                                                                                          | valuable basis for conducting a sensitivity experiment,                                                                                                                                                                                                                                                                                                                                                                                                                                                                                                                                                                                                                                                                                                                                                                        | Formatted: Font color: Auto, Not Raised b                                                    |
| 237
238                                                                                                                                   | valuable basis for conducting a sensitivity experiment,                                                                                                                                                                                                                                                                                                                                                                                                                                                                                                                                                                                                                                                                                                                                                                        | Formatted: Font color: Auto, Not Raised I                                                    |
| 237
238
239                                                                                                                            | valuable basis for conducting a sensitivity experiment,
2.3 Multivariate linear regression analysis                                                                                                                                                                                                                                                                                                                                                                                                                                                                                                                                                                                                                                                                                                                         | Formatted: Font color: Auto, Not Raised l                                                    |
| 237
238
239
240                                                                                                                     | valuable basis for conducting a sensitivity experiment,
2.3 Multivariate linear regression analysis
In this study we did a multivariate regression analysis of modeled monthly IEPOX-SOA (μg/m 3 )                                                                                                                                                                                                                                                                                                                                                                                                                                                                                                                                                                                                     | Formatted: Font color: Auto, Not Raised l                                                    |
| 237
238
239
240
241                                                                                                              | valuable basis for conducting a sensitivity experiment,
2.3 Multivariate linear regression analysis
In this study we did a multivariate regression analysis of modeled monthly IEPOX-SOA ( $\mu$ g/m 3 )
against modeled sulfate aerosol ( $\mu$ g/m 3 ), aerosol acidity $a_{H^+}$ (mol/L) and isoprene emission                                                                                                                                                                                                                                                                                                                                                                                                                                                                        | Formatted: Font color: Auto, Not Raised I                                                    |
| 237
238
239
240
241
242                                                                                                       | valuable basis for conducting a sensitivity experiment,
2.3 Multivariate linear regression analysis
In this study we did a multivariate regression analysis of modeled monthly IEPOX-SOA ( $\mu$ g/m 3 )
against modeled sulfate aerosol ( $\mu$ g/m 3 ), aerosol acidity $a_{H^+}$ (mol/L) and isoprene emission
( ISOP emis mg/m 2 /hr):                                                                                                                                                                                                                                                                                                                                                                                                               | Formatted: Font color: Auto, Not Raised I                                                    |
|  <li>237</li> <li>238</li> <li>239</li> <li>240</li> <li>241</li> <li>242</li> <li>243</li>                                         | valuable basis for conducting a sensitivity experiment,
2.3 Multivariate linear regression analysis
In this study we did a multivariate regression analysis of modeled monthly IEPOX-SOA ( $\mu$ g/m 3 )
against modeled sulfate aerosol ( $\mu$ g/m 3 ), aerosol acidity $a_{H^+}$ (mol/L) and isoprene emission
( ISOPemis mg/m 2 /hr):
IEPOX-SOA = $\beta_1 \times sulfate + \beta_2 \times a_{H^+} + \beta_3 \times ISOP_{emis} + b$                                                                                                                                                                                                                                                                                                       | Formatted: Font color: Auto, Not Raised I                                                    |
| 237

244                                                                                         | valuable basis for conducting a sensitivity experiment,
2.3 Multivariate linear regression analysis
In this study we did a multivariate regression analysis of modeled monthly IEPOX-SOA ( $\mu$ g/m 3 )
against modeled sulfate aerosol ( $\mu$ g/m 3 ), aerosol acidity $a_{H^+}$ (mol/L) and isoprene emission
( $ISOP_{emis}$ mg/m 2 /hr):
$IEPOX-SOA = \beta_1 \times sulfate + \beta_2 \times a_{H^+} + \beta_3 \times ISOP_{emis} + b$
Mean values have been subtracted from all variables, which are then divided by standard                                                                                                                                                                                                                                | Formatted: Font color: Auto, Not Raised I                                                    |
| 237

245                                                                                  | valuable basis for conducting a sensitivity experiment,
2.3 Multivariate linear regression analysis
In this study we did a multivariate regression analysis of modeled monthly IEPOX-SOA ( $\mu$ g/m 3 )
against modeled sulfate aerosol ( $\mu$ g/m 3 ), aerosol acidity $a_{H^+}$ (mol/L) and isoprene emission
( $ISOP_{emis}$ mg/m 2 /hr):
$IEPOX-SOA = \beta_1 \times sulfate + \beta_2 \times a_{H^+} + \beta_3 \times ISOP_{emis} + b$
Mean values have been subtracted from all variables, which are then divided by standard
deviations. $\beta_1$ , $\beta_2$ and $\beta_3$ are standardized partial regression coefficients associated with sulfate                                                                                                    | Formatted: Font color: Auto, Not Raised I                                                    |
|  <li>237</li> <li>238</li> <li>239</li> <li>240</li> <li>241</li> <li>242</li> <li>243</li> <li>244</li> <li>245</li> <li>246</li>  | valuable basis for conducting a sensitivity experiment,
2.3 Multivariate linear regression analysis
In this study we did a multivariate regression analysis of modeled monthly IEPOX-SOA ( $\mu$ g/m 3 )
against modeled sulfate aerosol ( $\mu$ g/m 3 ), aerosol acidity $a_{H^+}$ (mol/L) and isoprene emission
( $ISOP_{emis}$ mg/m 2 /hr):
$IEPOX-SOA = \beta_1 \times sulfate + \beta_2 \times a_{H^+} + \beta_3 \times ISOP_{emis} + b$
Mean values have been subtracted from all variables, which are then divided by standard
deviations. $\beta_1$ , $\beta_2$ and $\beta_3$ are standardized partial regression coefficients associated with sulfate
aerosol, $a_{H^+}$ and isoprene emission, and can be directly compared to evaluate the relative | Formatted: Font color: Auto, Not Raised I                                                    |

importance of the three variables. We apply the regression analysis using monthly data within 247

sions are ~52% ause the prior sions, in particular over the Central the prior y, southern lorth Carolina

by / Lowered by

256 different time frames (2000-2013, 2000-2004, 2005-2008 and 2009-2013 as in Table S1) to

257 determine the evolving importance of variables.

258

259

**260 **3. Results**

**261 3.1 Long-term trend and month-to-month variability (MMV) of OA**

262 In the southeast US, observations from the IMPROVE and SEARCH network both show a

263 reduction in summertime surface OA concentration from 2000 to 2013 (Figure 1). Observational

264 results are averaged using 21 IMPROVE sites and 3 SEARCH sites within the southeast US. OA

265 concentration averaged over June-July-August (JJA) 2000-2013 is 4.2 µg/m3 from the

266 IMPROVE sites, and 5.7  $\mu$ g/m3 from SEARCH sites. A similar ~30% summertime low bias on

267 the IMPROVE sites was documented by Kim et al. (2015) compared to the SEARCH sites,

268 which is thought to be due to evaporation of OA from the filters after collection, as the

269 IMPROVE filters stay several days on site after sampling and are shipped without refrigeration,

270 while the SEARCH filters are analyzed in-situ. Despite different magnitudes, OA from the two

271 networks demonstrate similar trends and interannual variability. The 2000-2013 trend of JJA OA

272 mass is -1.7%/year for IMPROVE and -1.9%/year for SEARCH. Compared to the slow decrease

- 273 in OA, a faster declining trend is found for sulfate from IMPROVE (-6.9%/year) and SEARCH
- (-6.7%/year) for the same period (Figure 2).
- 275

276 Compared to the observations, the default GEOS-Chem model predicts a steeper decreasing

trend of OA mass during 2000-2013 (Figure 1). Modeling results are averaged over the domain

278 [29°~37°N, 74°~96°W] excluding ocean grid cells (Figure S1). The 2000-2013 JJA-averaged

| 279                                                                                                                | OA from the default model is 6.7 $\mu$ g/m 3 , higher than OA from IMPROVE and SEARCH.                                                                                                                                                                                                                                                                                                                                                                                                                                                                                                                                                                                                                                                                   |
|--------------------------------------------------------------------------------------------------------------------|---------------------------------------------------------------------------------------------------------------------------------------------------------------------------------------------------------------------------------------------------------------------------------------------------------------------------------------------------------------------------------------------------------------------------------------------------------------------------------------------------------------------------------------------------------------------------------------------------------------------------------------------------------------------------------------------------------------------------------------------------------------------|
| 280                                                                                                                | Modeled total OA mass decreases at a rate of 4.9%/year, about 1.9 (1.6) times faster than                                                                                                                                                                                                                                                                                                                                                                                                                                                                                                                                                                                                                                                                           |
| 281                                                                                                                | IMPROVE (SEARCH) OA (student's t-test p<0.001). By sampling the model results at the                                                                                                                                                                                                                                                                                                                                                                                                                                                                                                                                                                                                                                                                                |
| 282                                                                                                                | locations of the IMPROVE and SEARCH sites, the modeled summertime OA has an average of                                                                                                                                                                                                                                                                                                                                                                                                                                                                                                                                                                                                                                                                              |
| 283                                                                                                                | $6.9 \mu g/m^3$ and a trend of 5.0%/year, similar to the model results averaged over the whole                                                                                                                                                                                                                                                                                                                                                                                                                                                                                                                                                                                                                                                                      |
| 284                                                                                                                | southeast US domain. For simplicity, we show only the domain-averaged model results in all                                                                                                                                                                                                                                                                                                                                                                                                                                                                                                                                                                                                                                                                          |
| 285                                                                                                                | figures and analysis. The strong reduction in total OA mass is dominated by aqueous SOA,                                                                                                                                                                                                                                                                                                                                                                                                                                                                                                                                                                                                                                                                            |
| 286                                                                                                                | especially through reactive uptake of IEPOX, with no decreasing trend in other components                                                                                                                                                                                                                                                                                                                                                                                                                                                                                                                                                                                                                                                                           |
| 287                                                                                                                | (Figure 1). The contribution of IEPOX-SOA to total OA mass decreases from 61% in the early                                                                                                                                                                                                                                                                                                                                                                                                                                                                                                                                                                                                                                                                          |
| 288                                                                                                                | 2000s to 28% in 2013. The simulated IEPOX-SOA in 2013 compares well with previous field                                                                                                                                                                                                                                                                                                                                                                                                                                                                                                                                                                                                                                                                             |
| 289                                                                                                                | studies which suggested that IEPOX-SOA contributed to 18~40% in southeast US sites in                                                                                                                                                                                                                                                                                                                                                                                                                                                                                                                                                                                                                                                                               |
| 290                                                                                                                | summer 2013 (Budisulistiorini et al., 2016; Xu et al., 2015b).                                                                                                                                                                                                                                                                                                                                                                                                                                                                                                                                                                                                                                                                                                      |
| 291                                                                                                                |                                                                                                                                                                                                                                                                                                                                                                                                                                                                                                                                                                                                                                                                                                                                                                     |
|                                                                                                                    |                                                                                                                                                                                                                                                                                                                                                                                                                                                                                                                                                                                                                                                                                                                                                                     |
| 292                                                                                                                | A main constraint comes from the MMV of OA in the southeast US. IMPROVE and SEARCH                                                                                                                                                                                                                                                                                                                                                                                                                                                                                                                                                                                                                                                                                  |
| 292
293                                                                                                         | A main constraint comes from the MMV of OA in the southeast US. IMPROVE and SEARCH
OA observations show little variability among June, July and August, despite large MMV of                                                                                                                                                                                                                                                                                                                                                                                                                                                                                                                                                                                     |
| 292
293
294                                                                                                  | A main constraint comes from the MMV of OA in the southeast US. IMPROVE and SEARCH
OA observations show little variability among June, July and August, despite large MMV of
sulfate in early 2000s (Figure 2A). We find similar behavior from another observation network,                                                                                                                                                                                                                                                                                                                                                                                                                                                                                   |
| 292
293
294
295                                                                                           | A main constraint comes from the MMV of OA in the southeast US. IMPROVE and SEARCH
OA observations show little variability among June, July and August, despite large MMV of
sulfate in early 2000s (Figure 2A). We find similar behavior from another observation network,
CSN. The discontinuity in OA trend in the CSN network is due to different protocols applied                                                                                                                                                                                                                                                                                                                                                                                    |
| 292
293
294
295
296                                                                                    | A main constraint comes from the MMV of OA in the southeast US. IMPROVE and SEARCH
OA observations show little variability among June, July and August, despite large MMV of
sulfate in early 2000s (Figure 2A). We find similar behavior from another observation network,
CSN. The discontinuity in OA trend in the CSN network is due to different protocols applied
(Figure S2). Within sites using the same protocol, there are no systematic monthly differences,                                                                                                                                                                                                                                                                                 |
| 292
293
294
295
296
297                                                                             | A main constraint comes from the MMV of OA in the southeast US. IMPROVE and SEARCH
OA observations show little variability among June, July and August, despite large MMV of
sulfate in early 2000s (Figure 2A). We find similar behavior from another observation network,
CSN. The discontinuity in OA trend in the CSN network is due to different protocols applied
(Figure S2). Within sites using the same protocol, there are no systematic monthly differences,
which agrees with IMPROVE and SEARCH. In contrast, modeled OA displays large MMV                                                                                                                                                                                             |
| <ol> <li>292</li> <li>293</li> <li>294</li> <li>295</li> <li>296</li> <li>297</li> <li>298</li> </ol>              | A main constraint comes from the MMV of OA in the southeast US. IMPROVE and SEARCH
OA observations show little variability among June, July and August, despite large MMV of
sulfate in early 2000s (Figure 2A). We find similar behavior from another observation network,
CSN. The discontinuity in OA trend in the CSN network is due to different protocols applied
(Figure S2). Within sites using the same protocol, there are no systematic monthly differences,
which agrees with IMPROVE and SEARCH. In contrast, modeled OA displays large MMV
between June, July and August from 2000 to 2008, where OA in July and August is 1~3 times of                                                                                             |
| <ol> <li>292</li> <li>293</li> <li>294</li> <li>295</li> <li>296</li> <li>297</li> <li>298</li> <li>299</li> </ol> | A main constraint comes from the MMV of OA in the southeast US. IMPROVE and SEARCH
OA observations show little variability among June, July and August, despite large MMV of
sulfate in early 2000s (Figure 2A). We find similar behavior from another observation network,
CSN. The discontinuity in OA trend in the CSN network is due to different protocols applied
(Figure S2). Within sites using the same protocol, there are no systematic monthly differences,
which agrees with IMPROVE and SEARCH. In contrast, modeled OA displays large MMV
between June, July and August from 2000 to 2008, where OA in July and August is 1~3 times of
June values (Figure 2A). Such large MMV is dominated by aqueous SOA, especially from the |

2 higher than the observed total OA (Figure 2). The other components including POA and dry 301

| 302 | SOA (including terpene-SOA and Anthropogenic SOA) formed through partitioning together                                                             |
|-----|----------------------------------------------------------------------------------------------------------------------------------------------------|
| 303 | have low concentrations and small MMV. The default model well captures the variability of                                                          |
| 304 | observed sulfate (Figure 2A), with an average of 3.8 $\mu g/m^3$ and a trend of -6.9%/year, as                                                     |
| 305 | compared to -6.9%/year (average concentration 4.2 $\mu g/m^3)$ from IMPROVE and -6.7%/year                                                         |
| 306 | (average concentration 4.3 $\mu$ g/m 3 ) from SEARCH.                                                                                   |
| 307 |                                                                                                                                                    |
| 308 | The large MMV in the model suggests a much stronger modeled OA dependence on sulfate than                                                          |
| 309 | observations. In 2000-2004, changes in modeled sulfate from June to July and/or August                                                             |
| 310 | correspond to large MMV of modeled OA mass. In contrast, little MMV is found in observed                                                           |
| 311 | OA mass during the same months despite large MMV in observed sulfate (Figure 2A). From a                                                           |
| 312 | linear regression analysis using all monthly data in 2000-2013, the OA-to-sulfate regression                                                       |
| 313 | slope is m =0.29 ( r 2 =0.25) from IMPROVE, m =0.51 ( r 2 =0.43) from SEARCH, and m =1.87 |
| 314 | $(r^2=0.57)$ from the default model, even though the default model well captures the magnitude,                                                    |
| 315 | trend, and monthly variability of observed sulfate. In summary, simulated total OA mass in the                                                     |
| 316 | standard GEOS-Chem model, dominated by IEPOX-SOA, has a steeper decreasing trend from                                                              |
| 317 | 2000 to 2013 than the observations, and has a large MMV indicating strong dependence on                                                            |
| 318 | sulfate.                                                                                                                                           |
| 319 |                                                                                                                                                    |

**320 3.2 What controls the modeled IEPOX-SOA variability?**

The strong dependence of IEPOX-SOA on sulfate is well-established by laboratory and field
work: wet sulfate particles provide the surface and volume of liquid media for IEPOX reactive
uptake (Budisulistiorini et al., 2017; Eddingsaas et al., 2010; Riva et al., 2016; Xu et al., 2015b,
2016), and serve as nucleophiles for nucleophilic addition to form organosulfates (Nguyen et al.,

| 325 | 2014; Surratt et al., 2007b). Sulfate (SO 4 2- ), together with ammonium (NH 4 + ), nitrate (NO 3 - ) and |
|-----|-----------------------------------------------------------------------------------------------------------------------------------------------------------------------------|
| 326 | other ions, regulates proton (H + ) activity $(a_{H+})$ that can catalyze the ring-opening of epoxide                                                            |
| 327 | group leading to the formation of IEPOX-SOA (Gaston et al., 2014; Pye et al., 2013; Surratt et                                                                              |
| 328 | al., 2007a). However, some recent studies suggest that IEPOX-SOA is not well correlated with                                                                                |
| 329 | aerosol acidity estimated from thermodynamic models (Budisulistiorini et al., 2015; Lin et al.,                                                                             |
| 330 | 2013; Xu et al., 2015b), although the lack of direct measurements of aerosol acidity may be a                                                                               |
| 331 | limitation. We use the GEOS-Chem model here to examine the simulated IEPOX-SOA                                                                                              |
| 332 | dependence on sulfate, aerosol acidity, and emissions of isoprene which produce IEPOX at high                                                                               |
| 333 | yields under low-NO x conditions (Paulot et al., 2009). Temperature impacts the formation of                                                              |
| 334 | IEPOX-SOA mainly through regulating isoprene emissions but does not influence partitioning as                                                                               |
| 335 | IEPOX-SOA is treated as non-volatile in GEOS-Chem. Therefore, temperature is not examined                                                                                   |
| 336 | as another driver in addition to isoprene emissions. We do not treat aerosol water as an                                                                                    |
| 337 | independent driver because the dilution effect of aerosol water is implicitly considered in the                                                                             |
| 338 | inorganic sulfate-ammonium-nitrate aerosol volume and acidity calculation, and studies have                                                                                 |
| 339 | shown that particle water is not a limiting factor unless the particle is purely dry (Nguyen et al.,                                                                        |
| 340 | 2014; Riva et al., 2016; Xu et al., 2015b) which is rare in summertime in the southeast US.                                                                                 |
| 341 |                                                                                                                                                                             |
| 342 | We find that the large MMV of OA in the model is mainly driven by sulfate concentrations and                                                                                |
| 343 | aerosol acidity. Figure 3 shows the standardized monthly surface IEPOX-SOA concentration,                                                                                   |
| 344 | sulfate concentration, aerosol H + activity and isoprene emission from the default model. For each                                                               |
| 345 | variable, the monthly gridded data has been first averaged over the southeast US. Then, we                                                                                  |
| 346 | calculate the one standard deviation of all monthly data (June, July and August data from 2000 to                                                                           |
| 347 |                                                                                                                                                                             |

[revised manuscript text omitted]

394increase in particle surface area. The sensitivity of  $\gamma_{IEPOX}$  to acidity has also been reduced395especially during the early 2000s (Figure 4A). The CT simulation reduces the southeast US JJA-396averaged IEPOX-SOA concentrations by  $0.3 \sim 1.8 \ \mu g/m^3$  (Figure 4C).

397

**398 3.3.2 NH3 emissions and aerosol acidity**

399 Second, recent studies present contradictory results and explanations on the long-term trend of 400 aerosol acidity in the southeast US (Pye et al., 2020; Silvern et al., 2017; Weber et al., 2016). In 401 this study, we show that the decreasing trend of aerosol acidity from the standard GEOS-Chem 402 model is mainly caused by high acidity in August before 2008, which corresponds to insufficient 403 NH3 emissions in high sulfate environments. The NEI11v1 inventory is used in the default 404 configuration, in which NH3 emissions in June and July are 30% higher than in August (Figure 405 S4), but not all NH3 emission inventories agree with such pattern (Paulot et al., 2014). We did a 406 sensitivity test ('CT newNH3') replacing the default US NH3 emissions from NEI11v1 by a new 407 NH3 emission product derived from CrIS satellite observations, which has higher emissions and 408 smaller MMV among June, July and August (Figure S4). In the 'CT newNH3' simulation, the 409 resulting simulated aerosol acidity is substantially changed in 2000-2008 (Figure 4B). The high 410 acidity  $(a_{H+}=0.55\sim0.9 \text{ mol/L})$  in August has been reduced to around 0.2 mol/L and is much 411 closer to June and July values (Figure 3). The results suggest that the fine particles in the 412 southeast US are within a regime where the acidity ( $a_{H+}$  in units of mol/L) is sensitive to NH3 413 emissions relative to sulfate concentration, though corresponding pH changes are small (pH 414 within 0.5~1.5, Figure S4). Small changes in NH3 may lead to large changes in  $a_{H+}$  especially 415 when sulfate concentrations are high, resulting in high month-to-month variability of the IEPOX 416 uptake. After updating the NH3 emissions using the satellite-based estimates, the model

| 41/ | simulates a much more stable trend in aerosol acidity from 2000 to 2015 (Figure 4B), consistent                       |                   |
|-----|-----------------------------------------------------------------------------------------------------------------------|-------------------|
| 418 | with recent thermodynamic modeling studies that suggested steady aerosol acidity despite large                        |                   |
| 419 | reductions in observed sulfate (Pye et al., 2020; Weber et al., 2016).                                                |                   |
| 420 |                                                                                                                       |                   |
| 421 | Due to the high uncertainty associated with the derived NH3 emission product and acidity                              |                   |
| 422 | calculation (Guo et al., 2015, 2018; Silvern et al., 2017; Song et al., 2018; Tao and Murphy,                         |                   |
| 423 | 2019), we conducted another simulation 'CT_H01' that fix $a_{H^+}$ level at 0.1 mol/L when                            |                   |
| 424 | calculating IEPOX uptake rate, corresponding to the predicted $a_{H+}$ value (constrained by                   | Deleted: observed |
| 425 | observations) during the 2013 SOAS campaign (Weber et al., 2016). The two simulations,                                |                   |
| 426 | $CT_newNH_3$ and $CT_H01$ , yield similar long-term trends of IEPOX-SOA in the southeast US                           |                   |
| 427 | (Figure S5), and they agree better with the long-term surface OA measurements from IMPROVE                            |                   |
| 428 | and SEARCH than the default model (Figure 4C and 4D). For the SOAS2013 campaign, the                                  |                   |
| 429 | CT_H01 scheme simulates an average IEPOX-SOA concentration of 0.74 $\mu$ g/m 3 , similar to 0.81           |                   |
| 430 | $\mu g/m^3$ in the default model, and agrees well with the two independent Aerosol Mass                               |                   |
| 431 | Spectrometer measurements (0.97 $\mu g/m^3$ from obs_GT and 0.68 $\mu g/m^3$ from obs_CU, see daily                   |                   |
| 432 | time series in Figure S $\underline{0}$ ). The CT_newNH 3 scheme simulates an average IEPOX-SOA            |                   |
| 433 | concentration of 0.34 $\mu$ g/m 3 , lower than the observation and the other models by a factor of >2,     |                   |
| 434 | due to both the simplified coating effect and small aerosol $a_{H^+}$ values ( $a_{H^+} < 0.1 \text{ mol/L}$ , Figure |                   |
| 435 | 4B). In general, the fixed acidity in the CT_H01 simulation well captures the measured IEPOX-                         |                   |
| 436 | SOA from the SOAS2013 campaign (Figure S6), and improves the modeled total OA mass                                    |                   |
| 437 | relative to the observations: The modeled long-term decreasing rate of JJA-average OA from                            |                   |
| 438 | 2000 to 2013 has been reduced from 4.9%/year to 3.2%/year, better compared to the IMPROVE                             |                   |

agal agidity from 2000 to 2012 (Eiguna

1.1

1 .

417

(1D)

[revised manuscript text omitted]

- 769
- Li, Y., Day, D. A., Stark, H., Jimenez, J. L. and Shiraiwa, M.: Predictions of the glass transition 770 temperature and viscosity of organic aerosols from volatility distributions, Atmos. Chem. Phys.,
- 20(13), 8103-8122, doi:10.5194/acp-20-8103-2020, 2020. 771
- 772 Liggio, J., Li, S. M. and McLaren, R.: Reactive uptake of glyoxal by particulate matter, J.
- Geophys. Res. D Atmos., 110(10), 1-13, doi:10.1029/2004JD005113, 2005. 773
- 774 Lin, Y. H., Knipping, E. M., Edgerton, E. S., Shaw, S. L. and Surratt, J. D.: Investigating the
- 775 influences of SO2 and NH3 levels on isoprene-derived secondary organic aerosol formation
- 776 using conditional sampling approaches, Atmos. Chem. Phys., 13(16), 8457-8470,
- 777 doi:10.5194/acp-13-8457-2013, 2013.
- 778 Lopez-Hilfiker, F. D., Mohr, C., D'Ambro, E. L., Lutz, A., Riedel, T. P., Gaston, C. J., Iyer, S.,
- Zhang, Z., Gold, A., Surratt, J. D., Lee, B. H., Kurten, T., Hu, W. W., Jimenez, J., Hallquist, M. 779
- 780 and Thornton, J. A.: Molecular Composition and Volatility of Organic Aerosol in the
- 781 Southeastern U.S.: Implications for IEPOX Derived SOA, Environ. Sci. Technol., 50(5), 2200-782 2209, doi:10.1021/acs.est.5b04769, 2016.
- 783 Malm, W. C., Schichtel, B. A., Hand, J. L. and Collett, J. L.: Concurrent Temporal and Spatial
- 784 Trends in Sulfate and Organic Mass Concentrations Measured in the IMPROVE Monitoring
- 785 Program, J. Geophys, Res. Atmos., 122(19), 10462-10476, doi:10.1002/2017JD026865, 2017.
- 786 Mao, J., Paulot, F., Jacob, D. J., Cohen, R. C., Crounse, J. D., Wennberg, P. O., Keller, C. A.,
- 787 Hudman, R. C., Barkley, M. P. and Horowitz, L. W.: Ozone and organic nitrates over the eastern 788 United States: Sensitivity to isoprene chemistry, J. Geophys. Res. Atmos., 118(19), 11256-
- 789 11268, doi:10.1002/jgrd.50817, 2013.
- 790 Mao, J., Carlton, A., Cohen, R. C., Brune, W. H., Brown, S. S., Wolfe, G. M., Jimenez, J. L.,
- 791 Pye, H. O. T., Lee Ng, N., Xu, L., Faye McNeill, V., Tsigaridis, K., McDonald, B. C., Warneke,
- 792 C., Guenther, A., Alvarado, M. J., De Gouw, J., Mickley, L. J., Leibensperger, E. M., Mathur,
- 793 R., Nolte, C. G., Portmann, R. W., Unger, N., Tosca, M. and Horowitz, L. W.: Southeast
- 794 Atmosphere Studies: Learning from model-observation syntheses, Atmos. Chem. Phys., 18(4),
- 795 2615-2651, doi:10.5194/acp-18-2615-2018, 2018.
- 796 Marais, E. A., Jacob, D. J., Jimenez, J. L., Campuzano-Jost, P., Day, D. A., Hu, W., Krechmer,
- 797 J., Zhu, L., Kim, P. S., Miller, C. C., Fisher, J. A., Travis, K., Yu, K., Hanisco, T. F., Wolfe, G.
- 798 M., Arkinson, H. L., Pve, H. O. T., Froyd, K. D., Liao, J. and McNeill, V. F.: Aqueous-phase
- 799 mechanism for secondary organic aerosol formation from isoprene: Application to the Southeast
- 800 United States and co-benefit of SO2 emission controls, Atmos. Chem. Phys., 16, 1603-1618,

- 801 doi:10.5194/acp-16-1603-2016, 2016.
- 802 Marais, E. A., Jacob, D. J., Turner, J. R. and Mickley, L. J.: Evidence of 1991–2013 decrease of
- 803 biogenic secondary organic aerosol in response to SO2 emission controls, Environ. Sci.
- 804 Technol., 12, doi:https://doi.org/10.1088/1748-9326/aa69c8, 2017.
- 805 McNeill, V. F., Woo, J. L., Kim, D. D., Schwier, A. N., Wannell, N. J., Sumner, A. J. and
- 806 Barakat, J. M.: Aqueous-phase secondary organic aerosol and organosulfate formation in
- atmospheric aerosols: A modeling study, Environ. Sci. Technol., 46(15), 8075–8081,
- 808 doi:10.1021/es3002986, 2012.
- 809 Murphy, D. M., Cziczo, D. J., Froyd, K. D., Hudson, P. K., Matthew, B. M., Middlebrook, A.
- 810 M., Peltier, R. E., Sullivan, A., Thomson, D. S. and Weber, R. J.: Single-peptide mass
- 811 spectrometry of tropospheric aerosol particles, J. Geophys. Res. Atmos., 111(23), 1–15,
- 812 doi:10.1029/2006JD007340, 2006.
- 813 Ng, N. L., Brown, S. S., Archibald, A. T., Atlas, E., Cohen, R. C., Crowley, J. N., Day, D. A.,
- 814 Donahue, N. M., Fry, J. L., Fuchs, H., Griffin, R. J., Guzman, M. I., Herrmann, H., Hodzic, A.,
- 815 Iinuma, Y., Kiendler-Scharr, A., Lee, B. H., Luecken, D. J., Mao, J., McLaren, R., Mutzel, A.,
- 816 Osthoff, H. D., Ouyang, B., Picquet-Varrault, B., Platt, U., Pye, H. O. T., Rudich, Y., Schwantes,
- 817 R. H., Shiraiwa, M., Stutz, J., Thornton, J. A., Tilgner, A., Williams, B. J. and Zaveri, R. A.:
- 818 Nitrate radicals and biogenic volatile organic compounds: Oxidation, mechanisms, and organic
- 819 aerosol, Atmos. Chem. Phys., 17(3), 2103–2162, doi:10.5194/acp-17-2103-2017, 2017.
- 820 Nguyen, T. B., Coggon, M. M., Bates, K. H., Zhang, X., Schwantes, R. H., Schilling, K. A.,
- 821 Loza, C. L., Flagan, R. C., Wennberg, P. O. and Seinfeld, J. H.: Organic aerosol formation from
- the reactive uptake of isoprene epoxydiols (IEPOX) onto non-acidified inorganic seeds, Atmos.
  Chem. Phys., 14(7), 3497–3510, doi:10.5194/acp-14-3497-2014, 2014.
- Pai, S., Heald, C., Pierce, J., Farina, S., Marais, E., Jimenez, J., Campuzano-Jost, P., Nault, B.,
- Middlebrook, A., Coe, H., Shilling, J., Bahreini, R., Dingle, J. and Vu, K.: An evaluation of
- global organic aerosol schemes using airborne observations, Atmos. Chem. Phys., (20), 2637–
- global organic deleter belenes asing anothe observations, rands: chem. Phys., (20)
   2665, doi:10.5194/acp-2019-331, 2020.
- Pankow, J. F.: An Absorption-Model of the Gas Aerosol Partitioning Involved in the Formation
  of Secondary Organic Aerosol, Atmos. Environ., 28(2), 189–193,
- 830 doi:10.1016/j.atmosenv.2007.10.060, 1994.
- 831 Paulot, F., Crounse, J. D., Kjaergaard, H. G., Kürten, A., Clair, J. M. S., Seinfeld, J. H. and
- 832 Wennberg, P. O.: Unexpected Epoxide Formation in the Gas-Phase Photooxidation of Isoprene,
- 833 Science (80-. )., 325, 730–734, doi:10.1126/science.1174251, 2009.
- 834 Paulot, F., Jacob, D. J., Pinder, R. W., Bash, J. O., Travis, K. R. and Henze, D. K.: Ammonia
- 835 emissions in the United States, European Union, and China derived by high-resolution inversion
- 836 of ammonium wet deposition data: Interpretation with a new agricultural emissions inventory
- 837 (MASAGE\_NH3), J. Geophys. Res. Atmos., 119, 4343–4364, doi:doi:10.1002/2013JD021130,
   838 2014.
- 839 Presto, A. a., Huff Hartz, K. E. and Donahue, N. M.: Secondary organic aerosol production from
- terpene ozonolysis. 2. Effect of NOx concentration, Environ. Sci. Technol., 39(18), 7046–7054,
  doi:10.1021/es050400s, 2005.
- 041 doi.10.1021/cs050400s, 200.
- 842 Pye, H. O. T., Liao, H., Wu, S., Mickley, L. J., Jacob, D. J., Henze, D. J. and Seinfeld, J. H.:
- 843 Effect of changes in climate and emissions on future sulfate-nitrate-ammonium aerosol levels in
- the United States, J. Geophys. Res. Atmos., 114(1), 1–18, doi:10.1029/2008JD010701, 2009.
- 845 Pye, H. O. T., Chan, a. W. H., Barkley, M. P. and Seinfeld, J. H.: Global modeling of organic
- aerosol: The importance of reactive nitrogen (NOx and NO3), Atmos. Chem. Phys., 10, 11261-

[revised manuscript text omitted]